# A simple regulatory architecture allows learning the statistical structure of a changing environment

Stefan Landmann[1], Caroline M Holmes[2], Mikhail Tikhonov[3]*

[1]Institute of Physics, Carl von Ossietzky University of Oldenburg, Oldenburg, Germany; [2]Department of Physics, Princeton University, Princeton, United States; [3]Department of Physics, Center for Science and Engineering of Living Systems, Washington University in St. Louis, St. Louis, United States

**Abstract** Bacteria live in environments that are continuously fluctuating and changing. Exploiting any predictability of such fluctuations can lead to an increased fitness. On longer timescales, bacteria can 'learn' the structure of these fluctuations through evolution. However, on shorter timescales, inferring the statistics of the environment and acting upon this information would need to be accomplished by physiological mechanisms. Here, we use a model of metabolism to show that a simple generalization of a common regulatory motif (end-product inhibition) is sufficient both for learning continuous-valued features of the statistical structure of the environment and for translating this information into predictive behavior; moreover, it accomplishes these tasks near-optimally. We discuss plausible genetic circuits that could instantiate the mechanism we describe, including one similar to the architecture of two-component signaling, and argue that the key ingredients required for such predictive behavior are readily accessible to bacteria.

## Introduction

Organisms that live in changing environments evolve strategies to respond to the fluctuations. Many such adaptations are reactive, for example sensory systems that allow detecting changes when they occur and responding to them. However, adaptations can be not only reactive, but also predictive. For example, circadian clocks allow photosynthetic algae to reorganize their metabolism in preparation for the rising sun (*Bell-Pedersen et al., 2005*; *Husain et al., 2019*). Another example is the anticipatory behavior in *E. coli*, which allows it to prepare for the next environment under its normal cycling through the mammalian digestive tract (*Savageau, 1983*); similar behaviors have been observed in many species (*Tagkopoulos et al., 2008*; *Mitchell et al., 2009*).

All these behaviors effectively constitute predictions about a future environment: the organism improves its fitness by exploiting the regularities it 'learns' over the course of its evolution (*Mitchell and Lim, 2016*). Learning such regularities can be beneficial even if they are merely statistical in nature. A prime example is bet hedging: even if the environment changes stochastically and without warning, a population that learns the statistics of switching can improve its long-term fitness, for example, by adopting persistor phenotypes with appropriate probability (*Kussell and Leibler, 2005*; *Veening et al., 2008*). The seemingly limitless ingenuity of evolutionary trial-and-error makes it plausible that virtually any statistical structure of the environment that remains constant over an evolutionary timescale could, in principle, be learnt by an evolving system, and harnessed to improve its fitness (*Watson and Szathmáry, 2016*).

However, the statistical structure of the environment can itself change, and this change can be too quick to be learned by evolution (*Figure 1A*). For example, an organism might experience a period of stability followed by a period of large fluctuations, or an environment where two resources

*For correspondence:
tikhonov@wustl.edu

**Competing interests:** The authors declare that no competing interests exist.

**eLife digest** Associations inferred from previous experience can help an organism predict what might happen the next time it faces a similar situation. For example, it could anticipate the presence of certain resources based on a correlated environmental cue.

The complex neural circuitry of the brain allows such associations to be learned and unlearned quickly, certainly within the lifetime of an animal. In contrast, the sub-cellular regulatory circuits of bacteria are only capable of very simple information processing. Thus, in bacteria, the 'learning' of environmental patterns is believed to mostly occur by evolutionary mechanisms, over many generations.

Landmann et al. used computer simulations and a simple theoretical model to show that bacteria need not be limited by the slow speed of evolutionary trial and error. A basic regulatory circuit could, theoretically, allow a bacterium to learn subtle relationships between environmental factors within its lifetime. The essential components for this simulation can all be found in bacteria – including a large number of 'regulators', the molecules that control the rate of biochemical processes. And indeed, some organisms often have more of these biological actors than appears to be necessary. The results of Landmann et al. provide new hypothesis for how such seemingly 'superfluous' elements might actually be useful.

Knowing that a learning process is theoretically possible, experimental biologists could now test if it appears in nature. Placing bacteria in more realistic, fluctuating conditions instead of a typical stable laboratory environment could demonstrate the role of the extra regulators in helping the microorganisms to adapt by 'learning'.

are correlated, and then another where they are not. Note that there are two key timescales here – that of the fluctuations themselves (which we assume to be fast), and the slower timescale on which the *structure* of those fluctuations changes. One expects such scenarios to be common in an eco-evolutionary context. As an example, consider a bacterium in a small pool of water. Its immediate environment, shaped by local interactions, is fluctuating on the timescale at which the bacterium changes neighbors. The statistical properties of these fluctuations depend on the species composition of the pool. As such, the fast fluctuations are partially predictable, and learning their structure could help inform the fitness-optimizing strategy: a neighbor encountered in a recent past is likely to be seen again in the near future. However, these statistics change on an ecological timescale, and such learning would therefore need to be accomplished by physiological, rather than evolutionary, mechanisms.

On a physiological timescale, this problem is highly nontrivial: the organism would have to perform inference from prior observations, encode them in memory, and act upon this knowledge (*Figure 1B*). It is clear that solutions to this problem do exist: such behaviors, common in neural systems, can be implemented by neural-network-like architectures; and these known architectures can be translated into biochemical networks (*Hjelmfelt et al., 1991*; *Kobayashi, 2010*; *Fages et al., 2017*; *Katz and Springer, 2016*; *Katz et al., 2018*). But single-celled organisms operate in a severely hardware-limited regime rarely probed by neuroscience. Streamlined by evolution, bacterial genomes quickly shed any unused complexity. Whether we could expect learning-like behaviors from bacteria depends on whether useful networks could be simple enough to plausibly be beneficial.

Known examples of phenotypic memory, for example, when the response is mediated by a long-lived protein, can be interpreted as a simple form of learning (*Lambert et al., 2014*; *Hoffer et al., 2001*); circuits capable of adapting to the current mean of a fluctuating signal, as in bacterial chemotaxis (*Barkai and Leibler, 1997*), also belong in this category. Prior theory work has also proposed that simple genetic circuits could learn more subtle binary features, such as a (transient) presence or absence of a correlation between two signals (*Sorek et al., 2013*).

Here, we show that a simple generalization of a ubiquitous regulatory motif, the end-product inhibition, can learn, store, and 'act upon' the information on continuous-valued features such as timescales and correlations of environmental fluctuations, and moreover, can do so near-optimally. We

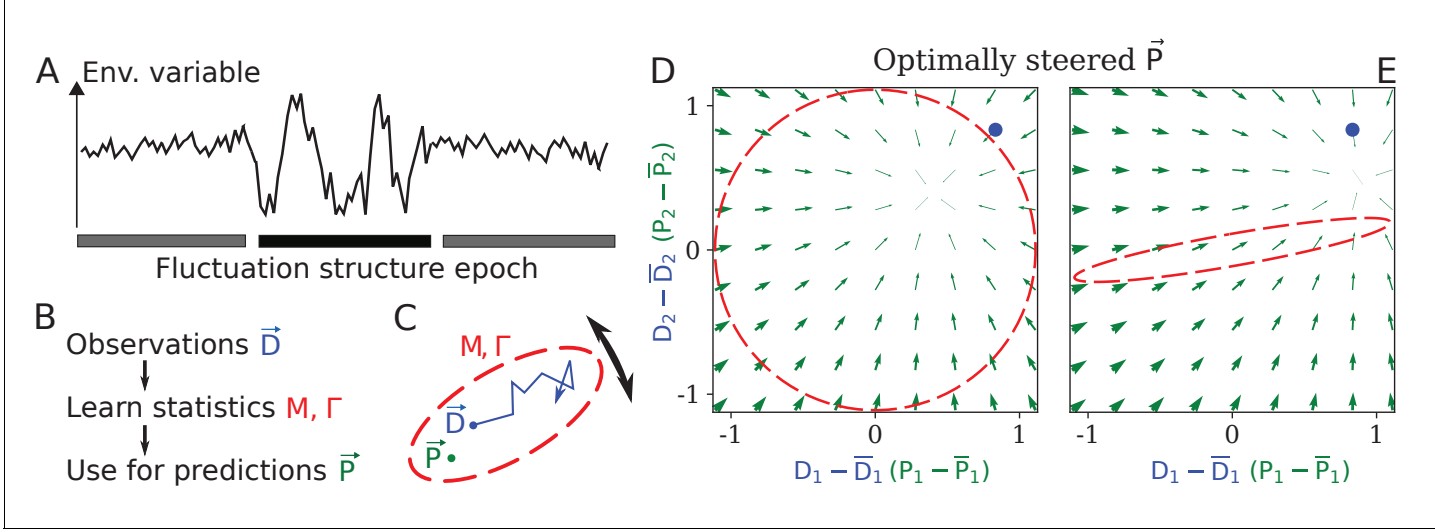

**Figure 1.** Learning environment statistics can benefit living systems, but is a difficult problem. (A) An environment is characterized not only by its current state, but also by its *fluctuation structure*, such as variances and correlations of fluctuating environmental parameters. In this work, we consider an environment undergoing epochs that differ in their fluctuation structure. Epochs are long compared to the physiological timescale, but switch faster than the evolutionary timescale. (B) The fluctuation structure can inform the fitness-maximizing strategy, but cannot be sensed directly. Instead, it would need to be learned from past observations, and used to inform future behavior. (C) To formalize the problem, we consider a situation where some internal physiological quantities $\vec{P}(t)$ must track fluctuating external factors $\vec{D}(t)$ undergoing a random walk. Since it is impossible to react instantaneously, $\vec{P}$ always lags behind $\vec{D}$. The dashed ellipse illustrates the fluctuation structure of $\vec{D}$ (encoded in parameters $M$ and $\Gamma$, see text), and changes on a slower timescale than the fluctuations of $\vec{D}$. (D, E) The optimal behavior in the two-dimensional version of our problem, under a constrained maximal rate of change $\|\dot{P}\|^2$. For a given current $\vec{D}$ (blue dot), the optimal control strategy would steer any current $\vec{P}$ (green arrows) toward the best guess of the future $\vec{D}$, which depends on the fluctuation structure (red ellipse: (D) fluctuations are uncorrelated and isotropic; (E) fluctuations have a preferred direction). The optimal strategy is derived using control theory (Appendix 1, section 'Control theory calculation').

identify the key ingredients giving rise to this behavior, and argue that their applicability is likely more general than the simple metabolically inspired example used here.

## Results

### The setup

For a simple model capturing some of the challenges of surviving in a fluctuating environment, consider a situation where some internal physiological quantities $\vec{P} = (P_1, \ldots, P_N)$ must track fluctuating external variables $\vec{D} = (D_1, \ldots, D_N)$. For example, the expression of a costly metabolic pathway would ideally track the availability of the relevant nutrient, or the solute concentration in the cytoplasm might track the osmolarity of the environment. In abstract terms, we describe these environmental pressures by the time-dependent $\vec{D}(t)$, and postulate that the organism fitness is determined by the average mismatch $-\sqrt{\langle \sum_{i=1}^{N} (P_i - D_i)^2 \rangle}$, a quantity we will henceforth call 'performance'. Here and below, angular brackets denote averaging over time.

In this simple model, a given static $\vec{D}$ clearly defines a unique optimal state $\vec{P}$; the regulatory challenge is entirely due to $\vec{D}$ being a fluctuating quantity. The challenges faced by real organisms are certainly vastly more rich: even in the static case, the optimal behavior may not be unique, or even well-defined (optimal under what constraints?); and in the dynamic case, the future state of the environment could be affected by past actions of the organism. These considerations can add layers of complexity to the problem, but our minimal model is sufficient to focus on the basic issues of sensing, learning and responding to changing fluctuation statistics of external factors.

If $\vec{D}$ changes sufficiently slowly, the organism can sense it and adapt $\vec{P}$ accordingly. We, instead, are interested in the regime of rapid fluctuations. When changes in $\vec{D}$ are too rapid for the organism

to match $\vec{P}$ to $\vec{D}$ exactly, it can rely on statistical structure. At the simplest level, the organism could match the mean, setting $\vec{P} \equiv \langle \vec{D} \rangle$. However, information on higher-order statistics, for example correlations between $D_1$ and $D_2$, can further inform the behavior and improve fitness.

To see this, in what follows, we will consider the minimal case of such structured fluctuations, namely a $N$-dimensional vector $\vec{D} = (D_1, \ldots, D_N)$ undergoing a random walk in a quadratic potential (the Ornstein—Uhlenbeck process):

$$\vec{D}(t + \Delta t) = \vec{D}(t) - M \cdot \left( \vec{D}(t) - \overline{\vec{D}} \right) \Delta t + \sqrt{2\Gamma\Delta t}\, \vec{\eta}, \tag{1}$$

with mean $\overline{\vec{D}}$, fluctuation strength $\Gamma$, independent Gaussian random variables $\vec{\eta}$ with zero mean and unit variance, and the matrix $M$ defining the potential.

In this system, the relevant 'fluctuation structure' is determined by $M$ and $\Gamma$. In one dimension, *Equation (1)* gives $D$ a variance of $\Gamma/M$. In two dimensions, denoting the eigenvalues of $M$ as $\lambda_{1,2}$, the stationary distribution of the fluctuating $\vec{D}$ is a Gaussian distribution with principal axes oriented along the eigenvectors of $M$, and standard deviations along these directions given by $\sqrt{\Gamma/\lambda_1}$ and $\sqrt{\Gamma/\lambda_2}$. Intuitively, we can think of the fluctuating $\vec{D}$ as filling out an ellipse (*Figure 1C*). Going forward, when we refer to learning fluctuation structure, we mean learning properties of $M$ and $\Gamma$.

If $M$ and $\Gamma$ are known, the optimal strategy minimizing $\langle (\vec{P} - \vec{D})^2 \rangle$, where $\vec{D}(t)$ is set by *Equation (1)*, can be computed exactly, as a function of the maximum allowed rate of change $\|\dot{P}\|^2$ (*Liberzon, 2011*). (If we do not constrain $\|\dot{P}\|^2$, the optimal behavior is of course $\vec{P} = \vec{D}$.) Briefly, the optimal behavior is to steer $\vec{P}$ toward the best guess of the expected future $\vec{D}$ (see Appendix 1, section 'Control theory calculation'). This best guess depends on the fluctuation structure, as illustrated by the comparison between *Figure 1D and E* for an isotropic and an anisotropic $M$.

However, in our problem, we will assume that $M$ and $\Gamma$ do not stay constant long enough to be learned by evolution, and thus are unknown to the system. In this regime, it is not clear that the behavior of an $M$- and $\Gamma$-optimized system is relevant. Nevertheless, we will describe a regulatory architecture consisting of common regulatory elements that will adapt its responsiveness to the fluctuation structure of its input ('learn'); for example, in the two-dimensional case, it will indeed develop the anisotropic response shown in *Figure 1E*. Moreover, we will find the steady-state performance of our architecture to be near-optimal, when compared to the theoretical ceiling of a system that knows $M$ and $\Gamma$ perfectly.

## Proposed architecture: end-product inhibition with an excess of regulators

The section above was intentionally very general. To discuss solutions available to cells, it is convenient to restrict the scope from this general formulation to a more specific metabolically-inspired case. From here onwards, let $D_i$ be the instantaneous demand in metabolite $x_i$ (determined by external factors), and $P_i$ be the rate at which the metabolite is produced, with both defined in units of metabolite concentration per unit time. The number of components of the vector $\vec{D}$ now has the meaning of the number of metabolites, and we will denote it as $N_x$. The cell needs to match $\vec{P}$ to $\vec{D}$ (or, equivalently, maintain the homeostasis of the internal metabolite concentrations $x_i$).

The simplest way to solve this problem is via feedback inhibition. Consider first the case of a single metabolite $x$. If an accumulation of $x$ inhibits its own synthesis, a decreased demand will automatically translate into a decreased production. For our purposes, we will model this scenario by placing the synthesis of metabolite $x$ under the control of a regulator $a$ (e.g. a transcription factor), which is, in turn, inhibited by $x$ (*Figure 2A*). For simplicity, we will measure regulator activity $a$ directly in units of equivalent production of $x$. The dynamics of this system, linearized for small fluctuations of metabolite concentration $x$, can be written in the following form (see Appendix 1, section 'Simple end-product inhibition'):

$$\dot{x} = P - D\frac{x}{x_0} \qquad \text{source-sink dynamics of metabolite } x \tag{2a}$$

$$P = aP_0 \qquad \text{definition of regulator activity } a \tag{2b}$$

$$\dot{a} = \frac{x_0 - x}{\lambda} \qquad \text{regulator activity inhibited by } x \tag{2c}$$

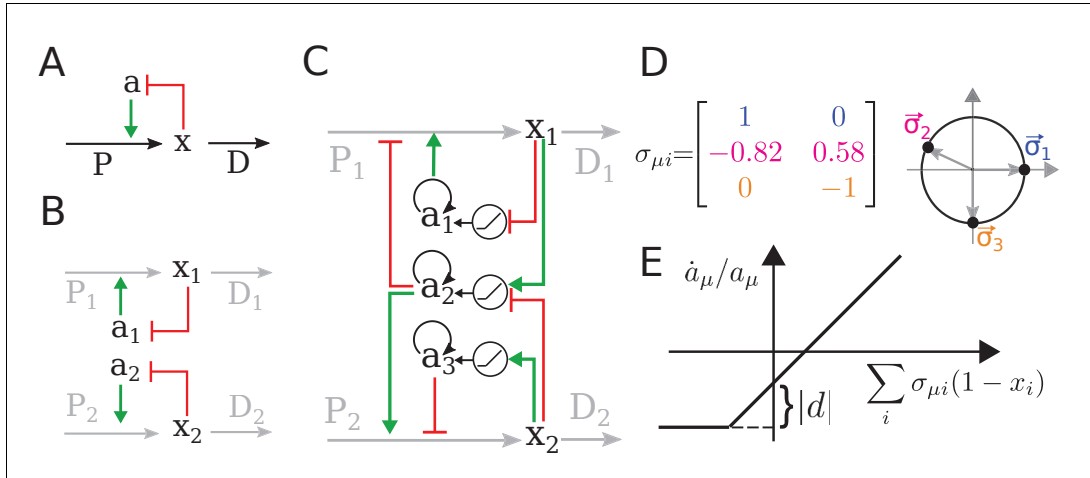

**Figure 2.** The regulatory architecture we consider is a simple generalization of end-product inhibition. (A) Simple end-product inhibition (SEPI) for one metabolite. Green arrows show activation, red arrows inhibition. (B) Natural extension of SEPI to several metabolites. (C) We consider regulatory architectures with more regulators than metabolites, with added self-activation (circular arrows) and a nonlinear activation/repression of regulators $a_\mu$ by the metabolite concentrations $x_i$ (pictograms in circles). (D) Visualizing a regulation matrix $\sigma_{\mu i}$ for two metabolites. In this example, the first regulator described by $\vec{\sigma}_1$ activates the production of $x_1$; the second inhibits $x_1$ and activates $x_2$. For simplicity, we choose vectors of unit length, which can be represented by a dot on the unit circle. This provides a convenient way to visualize a given regulatory architecture. (E) The nonlinear dependence of regulator activity dynamics $\dot{a}_\mu/a_\mu$ on metabolite concentrations $x_i$ in our model (see **Equation 4**).

Here, we introduced $P_0$ with dimension of production (concentration per time) to render $a$ dimensionless. In **Equation 2**c, $\lambda$ has the units of concentration $\times$ time, and setting $\lambda \equiv x_0 \tau_a$ defines a time scale for changes in regulator activity. Assuming the dynamics of metabolite concentrations $x$ are faster than regulatory processes, and choosing the units so that $x_0 = 1$ and $P_0 = 1$, we simplify the equations to:

$$
\begin{aligned}
x &= P/D \\
P &= a \\
\tau_a \dot{a} &= 1 - x.
\end{aligned}
\tag{3}
$$

We will refer to this architecture as simple end-product inhibition (SEPI). For two metabolites $\vec{x} = (x_1, x_2)$, the straightforward generalization is to have two independent copies of this circuit, with two regulators $a_1$, $a_2$ (**Figure 2B**). Denoting the number of regulators as $N_a$, we note that in the SEPI architecture, there are as many regulators as there are metabolites: $N_a = N_x$.

The architecture we will describe builds on this widely used regulatory motif, and relies on three added ingredients:

1. An excess of regulators: $N_a > N_x$;
2. Self-activation of regulators;
3. Nonlinear activation/repression of the regulators $a_\mu$ by the metabolite concentrations $x_i$.

Here and below, we use index $\mu$ for regulators ($\mu = 1 \ldots N_a$) and index $i$ for metabolites ($i = 1 \ldots N_x$).

These three ingredients, we claim, will be sufficient for the circuit to both learn higher order statistics and to use this information appropriately when matching the production to demand. It is important to emphasize that all three are readily accessible to cells. In fact, there are multiple ways to build regulatory circuits exhibiting the proposed behavior using common regulatory elements. To focus on the general mechanism rather than any one particular implementation, we will defer describing these example circuits until later in the text (Figure 6); here, we will consider a minimal modification of **Equation (3)** that contains the required ingredients:

$$x_i = P_i/D_i \tag{4a}$$

$$P_i = \Sigma_\mu \sigma_{\mu i} a_\mu \tag{4b}$$

$$\tau_a \dot{a}_\mu = a_\mu \max\left(d, \sum_i \sigma_{\mu i}(1 - x_i)\right) - \kappa a_\mu. \tag{4c}$$

This architecture bears a similarity to neural networks, and, as we will see, the familiar intuition about the value of extra 'hidden nodes' indeed holds. However, we caution the reader not to rely too heavily on this analogy. For example, here $\sigma_{\mu i}$ is a *constant* matrix describing how the activities of regulators $a_\mu$ control the synthesis of metabolites $x_i$.

For two metabolites ($N_x = 2$) as in **Figure 2C**, each regulator is summarized by a 2-component vector $\vec{\sigma}_\mu = (\sigma_{\mu 1}, \sigma_{\mu 2})$; its components can be of either sign (or zero) and specify how strongly the regulator $a_\mu$ is activating or repressing the synthesis of metabolite $x_i$. For simplicity, below, we will choose these vectors to be of unit length. Then, each regulator $\vec{\sigma}_\mu$ is fully characterized by an angle in the $(x_1, x_2)$ plane, which allows for a convenient visualization of the regulatory systems (**Figure 2D**). The $\sigma_{\mu i}$ defines the regulatory logic of our system and does not change with time. The parameter $d \leq 0$ allows us to tune the strength of the simple nonlinearity (**Figure 2E**); below we set $d = 0$ (strong nonlinearity) unless explicitly stated otherwise. As we will show later, the learning behavior is also observed for more realistic functions such as the Hill function, but the simple piece-wise linear form of **Equation (4)** will help us relate the observed behavior to specifically nonlinearity as opposed to, for example, cooperativity (the Hill parameter tunes both simultaneously). Finally, the parameter $\kappa$ reflects degradation and is assumed to be small: $\kappa \ll x_0$. Previously, for SEPI, it could be neglected, but here, it will matter due to the nonlinearity; for more details, see Appendix 1, section 'Simple end-product inhibition'. The parameters used in simulations are all listed in Appendix 1, section 'Parameters used in figures'.

Just like simple end-product inhibition in **Equation (3)**, the modified system **Equation (4)** will correctly adapt production to any static demand (see Appendix 1, section 'Adaptation to static demand'). In the following, we will show that the added ingredients also enable learning the structure of fluctuating environments. For this purpose, we expose our system to demands $D(t)$ with fixed means ($\overline{D}_i = 1$) but a changing fluctuation structure.

## The regulatory architecture described above outperforms simple end-product inhibition by learning environment statistics

To show that our system is able to adapt to different fluctuation structures, we probe it with changing environmental statistics, and show that it, first, learns these statistics, and, second, is able to make use of this information in its behavior.

For simplicity, we start with the 1-dimensional case (**Figure 3A–F**). In dimension $N_x = 1$, an excess of regulators means we have both an activator $a_+$ and a repressor $a_-$ for the production of $x$ (**Figure 3A**). This is reminiscent of paradoxical regulation (**Hart et al., 2012**). We probe our system with changing environmental statistics by exposing it to a demand $D(t)$ with an increasing variance (**Figure 3B,C**). As a reminder, here and below, the mean demand is fixed at 1.

Faced with a faster fluctuating input, our system upregulates both $a_+$ and $a_-$ while keeping $a_+ - a_-$ constant ($a_+ - a_- \approx \overline{D} = 1$; **Figure 3D**). In this way, the two activity levels $a_+$ and $a_-$ encode both the mean and the variance of fluctuations. Crucially, the system makes use of the information it stores: The increased regulator activities allow future changes in $P$ to be faster. The system's *responsiveness*, which we can define as $\mathcal{R} \equiv \frac{d\dot{P}}{dD}$, increases as $a_+ + a_-$ (**Figure 3E**; see also Appendix 1, section 'Defining the system's responsiveness'). As a result, as shown in **Figure 3F**, our system is able to perform approximately equally well (after adaptation time) in each environment, unlike a system like simple end-product inhibition, which is unable to adapt its sensitivity. In summary, **Figure 3D–F** show that the simple architecture of **Figure 3A** can not only learn the statistics of environment fluctuations, but also 'act upon this knowledge,' effectively performing both computations of **Figure 1B**.

The idea of learning the fluctuation structure is perhaps clearer in dimension $N_x = 2$, since the two demands can now be correlated with each other, and it seems intuitive that a system able to learn the typical direction of fluctuations (the angle $\alpha$ in **Figure 3H**) should be able to track the input better. Indeed, as we saw in **Figure 1D–E**, when environment fluctuations are anisotropic, the

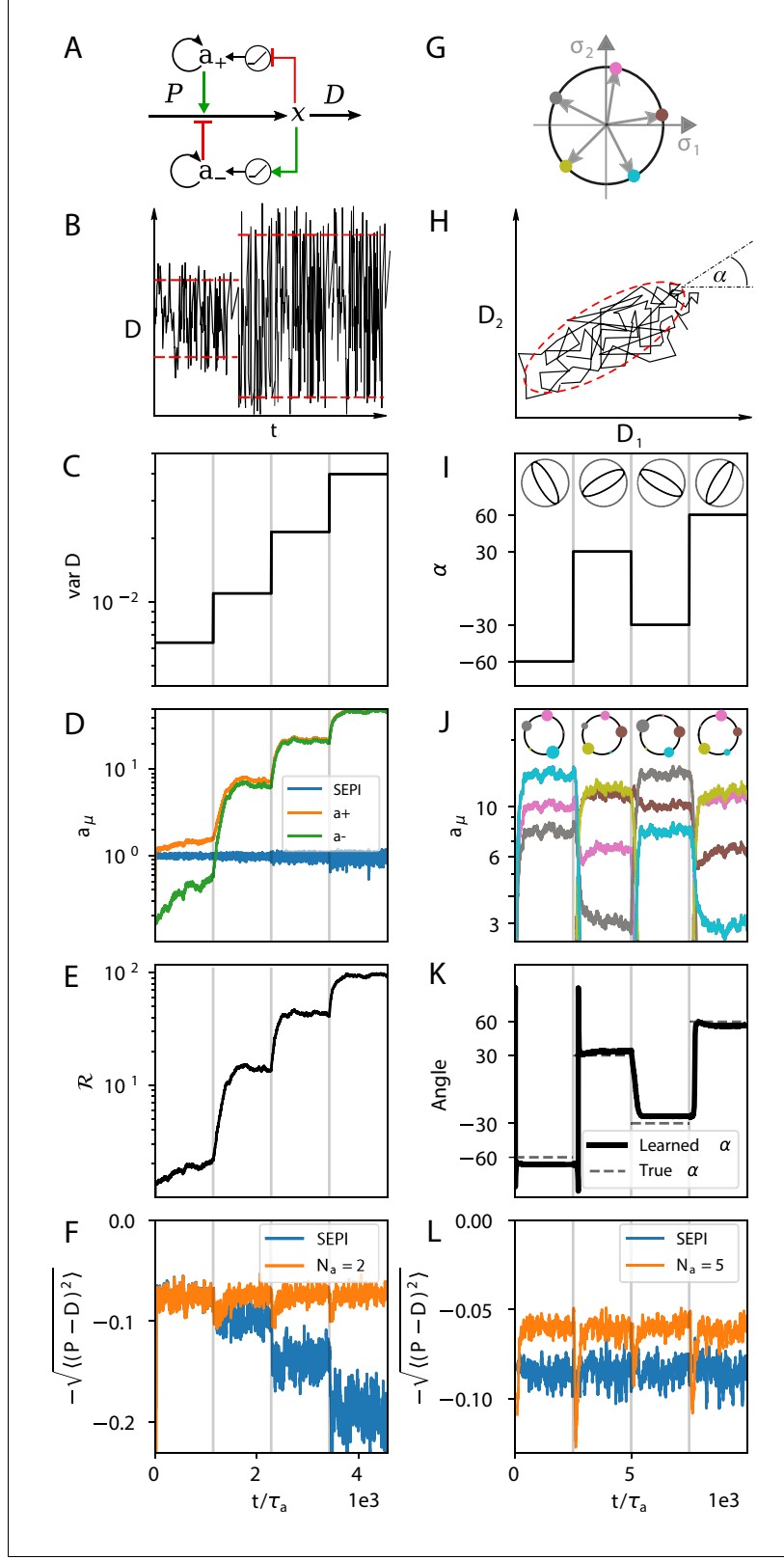

**Figure 3.** The regulatory architecture we consider successfully learns environment statistics, and outperforms simple end-product inhibition. Left column in one dimension, right column in two. (A) Regulation of a single metabolite $x$ with one activator $a_+$ and one repressor $a_-$. (B, C) The variance of $D$ is increased step-wise (by increasing $\Gamma$). (D) Regulator activities $a_\pm$ respond to the changing statistics of $\vec{D}$. For SEPI, the activity of its single

*Figure 3 continued on next page*

*Figure 3 continued*

regulator is unchanged. (E) Faced with larger fluctuations, our system becomes more responsive. (F) As fluctuations increase, SEPI performance drops, while the circuit of panel A retains its performance. (G) In the 2d case, we consider a system with $N_a = 5$ regulators; visualization as in *Figure 2D*. (H) Cartoon of correlated demands with a dominant fluctuation direction (angle α). (I) We use α to change the fluctuation structure of the input. (J) Regulator activities respond to the changing statistics of $\vec{D}$. Colors as in panel G. (K) The direction of largest responsiveness ('learned angle'; see text) tracks the α of the input. (L) The system able to learn the dominant direction of fluctuations outperforms the SEPI architecture, even if the timescale $\tau_a$ of SEPI is adjusted to match the faster responsiveness of the $N_a = 5$ system (see Appendix 1, section 'Parameters used in figures'). Panels B and H are cartoons.

responsiveness of a well-adapted strategy must be anisotropic as well: the preferred direction must elicit a stronger response. Mathematically, the responsiveness $\mathcal{R}$ is now a matrix $\mathcal{R}_{ij} = \frac{d\dot{P}_i}{dD_j}$, and for a well-adapted system we expect its eigenvectors to align with the principal directions of $M$. In *Figure 3G–L*, *Figure 4* and *Figure 5*, our discussion will focus on this two-dimensional case.

  *Figure 3G–L* show the behavior of our system (*Equation 4*) with $N_a = 5$ regulators (*Figure 3G*), exposed to an input structured as shown in *Figure 3H*, where we vary α. As represented pictorially in *Figure 3I*, we rotate the fluctuation structure matrix $M$ in *Equation (1)*, keeping its eigenvalues $\lambda_{1,2}$ fixed with $\sqrt{\lambda_1/\lambda_2} = 10$ (this fixes the ratio of major to minor semi-axis lengths).

  With $N_a = 5$ regulators, matching the mean value of $\vec{D}$ would leave $N_a - 2 = 3$ degrees of freedom that can be influenced by other parameters (such as variance in each dimension and correlation between different demands). And indeed, changing environment statistics induces strong changes in the regulator state adopted by the system, with regulators better aligned to the input fluctuations reaching higher expression (*Figure 3J*; note the diagrams at the top, where dot size reflects the activity reached by the corresponding regulator at the end of each epoch; compare to the diagrams

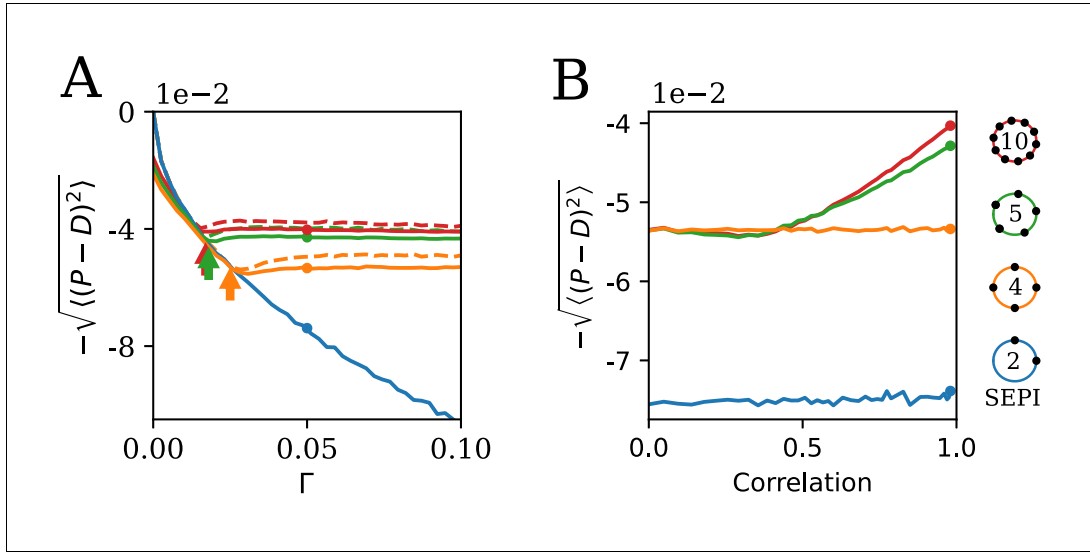

**Figure 4.** The ability to learn statistics is most useful when fluctuations are large and/or strongly correlated. (A) The performance of different circuits shown as a function of Γ, which scales the fluctuation magnitude (input is two-dimensional and correlated, angle $\alpha = \pi/4$, anisotropy $\sqrt{\lambda_1/\lambda_2} = 10$). Once the fluctuations become large enough to activate the learning mechanism, performance stabilizes; in contrast, the SEPI performance continues to decline. Arrows indicate the theoretical prediction for the threshold value of Γ; see Appendix 1, section 'The minimal Γ needed to initiate adaptation'. Dashed lines indicate the theoretical performance ceiling (calculated at equivalent Control Input Power, see text). (B) Comparison of circuit performance for inputs of the same variance, but different correlation strengths. $N_a = 4$ regulators arranged as shown can learn the variance but not correlation; the SEPI architecture is unable to adapt to either. Parameter Γ is held constant at 0.05; the marked points are identical to those highlighted in panel A (and correspond to fluctuation anisotropy $\sqrt{\lambda_1/\lambda_2} = 10$).

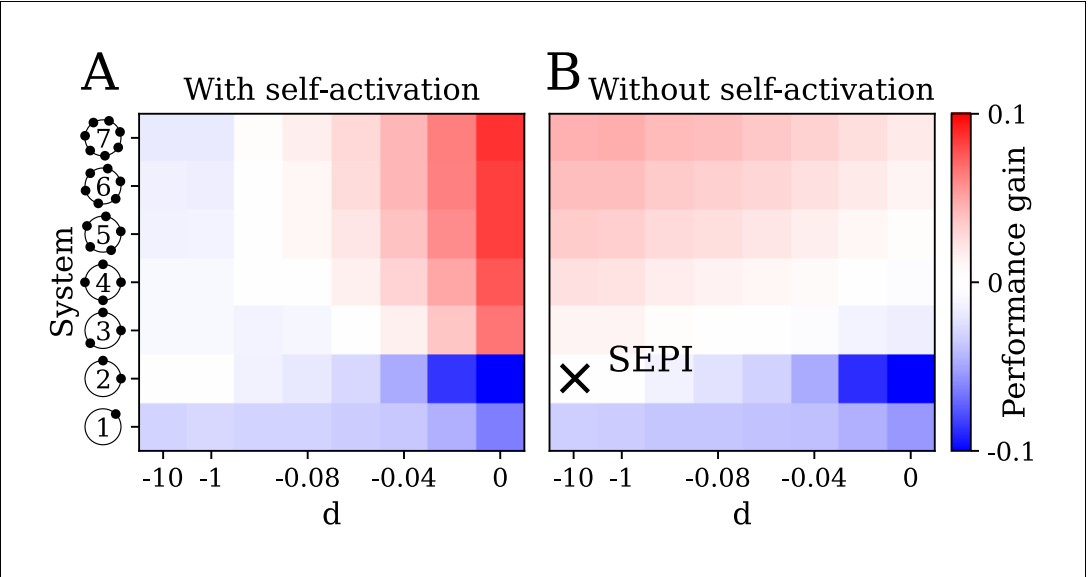

**Figure 5.** The key ingredients enabling learning are an excess of regulators, nonlinearity, and self-activation. (**A**) System performance in the two-dimensional case $N_x = 2$, shown as a function of the number of regulators $N_a$ (vertical axis) and the strength of nonlinearity $d$ (horizontal axis; $d = -10$ is indistinguishable from a linear system with $d = -\infty$). Color indicates performance gain relative to the SEPI architecture; performance is averaged over angle $\alpha$ (see **Figure 3H**). (**B**) Same as panel A, for a model without self-activation (see text). The SEPI-like architecture (linear with $N_a = 2$) is highlighted.

in **Figure 3I**). This activity pattern shapes the responsiveness matrix $\mathcal{R}$. **Figure 3K** plots the 'learned angle', defined as the direction of the dominant eigenvector of $\mathcal{R}$; we find that it tracks the stimulus angle. Finally, **Figure 3L** demonstrates that our architecture is able to make use of this learning, out-performing the SEPI system, whose responsiveness is isotropic and fixed.

## The performance is near-optimal

In the previous section, we have shown by example (**Figure 3**) that the proposed regulatory architecture can learn the statistics of the environment. We now characterize systematically the conditions under which learning improves performance and compare our system to the theoretical performance ceiling. Note that unlike the general statement that learning correlations improves performance, the *optimal* performance ceiling is necessarily specific to a given model of the environmental fluctuations. Nevertheless, this comparison is informative.

The fluctuation structure in our model is defined by $\Gamma$ and $M$. We first investigate the dependence of performance on $\Gamma$ (**Figure 4A**), exposing our system to a two-dimensional input structured as in **Figure 3H** with $\sqrt{\lambda_1/\lambda_2} = 10$ as before, $\alpha = \pi/4$, and a changing $\Gamma$.

Although the input is two-dimensional, changing $\Gamma$ scales the overall magnitude of fluctuations, and the behavior is analogous to the simpler one-dimensional example shown in the first column of **Figure 3**. At $\Gamma = 0$ (static input), and by extension, for $\Gamma$ finite but small, examining the steady state of **Equation (4)** shows that only $N_x = 2$ out of $N_a$ regulators can be active. In this regime, our system is essentially identical to SEPI—the extra regulators, though available, are inactive—and in fact performs slightly worse. This is because at nonzero $\kappa$, the steady state of **Equation (4)** is slightly offset from the ideal state $\langle x_i \rangle = 1$. (While this effect can be corrected, it is only relevant in the parameter regime where no learning occurs, so we chose to keep **Equation (4)** as simple as possible; for additional discussion, see Appendix 1, section 'Performance penalty from the degradation term').

When $\Gamma$ becomes sufficiently large, the first term in **Equation (4)** (proportional to the fluctuation size $\sqrt{\Gamma}$) for one of the inactive regulators finally exceeds, on average, the degradation term. At this point, the system enters the regime where the number of active regulators exceeds $N_x$, and its performance deviates from the SEPI curve. Beyond this point, further changes to the stimulus no longer affect performance, as our system is able to adapt its responsiveness to the changing fluctuation

magnitude (compare to *Figure 3F*). The threshold value of $\Gamma$ satisfies $\sqrt{\Gamma} \propto \kappa$; the proportionality coefficient of order 1 depends on the specific arrangement of regulators but can be estimated analytically (see Appendix 1, section 'The minimal $\Gamma$ needed to initiate adaptation'). The theoretically predicted deviation points are indicated with arrows, and are in agreement with the simulation results. When a regulator in the system is particularly well-aligned with the dominant direction of fluctuations, the deviation occurs sooner, explaining the better performance of our system when the regulators are more numerous.

To better assess the performance of our system, we compare it to the theoretical optimum derived from control theory, which we represent with dotted lines in *Figure 4A*. For given $M$ and $\Gamma$, the family of optimal behaviors is parameterized by Control Input Power (CIP), defined as $\int \|\dot{P}\|^2 \, dt$. If $\vec{P}$ could react infinitely fast, it would track $\vec{D}$ perfectly, but increasing response speed necessarily comes at a cost (of making more sensors, or more enzymes for faster synthesis / degradation of $x_i$); constraining the CIP is thus a proxy for specifying the maximum tolerable cost. In order to compare our system with the optimal family of solutions, we compute $\frac{1}{T}\int_0^T \|\dot{P}\|^2 \, dt$ of our system at each $\Gamma$ ($T$ is the simulation time), and compare to the performance of the optimally steered solution with a matched CIP; details of the calculation can be found in Appendix 1, section 'Control theory calculation'. *Figure 4A* demonstrates that the simple architecture we described not only benefits from matching its responsiveness to its input, but is in fact near-optimal when compared to *any* system of equivalent responsiveness.

It is important to note that for low $\Gamma$, the performance of the SEPI architecture also tracks the optimal curve. Throughout this work, our intention is not to demonstrate that SEPI is a 'poor' architecture. To the contrary, the surprising efficiency of SEPI has been noted before (*Goyal et al., 2010*; *Pavlov and Ehrenberg, 2013*), and *Figure 4* similarly shows that at its own CIP, its performance is virtually optimal. The advantage of our learning-capable architecture derives from its ability to increase responsiveness when necessary, in the correct direction. Our simplified treatment of the SEPI architecture is not a strawman we seek to dismiss, but an example of a system that exhibits no learning.

Having investigated the effect of fluctuation variance (changing $\Gamma$), we turn to the effect of their correlation. Up to now, we subjected our system to a strongly correlated two-dimensional input with anisotropy $\sqrt{\lambda_1/\lambda_2} = 10$ (illustrated, to scale, in *Figure 1E*). We will now consider a range of anisotropy values, down to anisotropy 1 (uncorrelated fluctuations, *Figure 1D*), keeping the variances of $D_1$ and $D_2$ constant, $\alpha = \pi/4$ as before, and $\Gamma = 0.05$.

The result is presented in *Figure 4B*. With $N_a = 5$ or larger, our system is able to take advantage of the correlation, assuming it is strong enough to activate the learning mechanism. (In fact, its performance can reach values that exceed the theoretical ceiling achievable by any system that assumes the two dimensions of $\vec{D}$ to be independent, and thus *must* be exploiting the correlation in its inputs; see Appendix 1, section 'The system makes use of correlations in the input' and *Appendix 1—figure 1*). For $N_a = 4$, the performance curve remains flat. This is because the four regulators are arranged as two independent copies of the system shown in *Figure 3A* (one $\{a_+, a_-\}$ pair for each of the two inputs); this architecture can take advantage of the learned variance, but not the correlation. Finally, the SEPI architecture can adapt to neither variance nor correlation; its performance curve is also flat, but is lower. As expected, the advantage of our architecture manifests itself in environments with periods of large and/or strongly correlated fluctuations.

## The behavior is generalizable

The model described above was a proof of principle, showing that simple regulatory circuits could learn the fluctuation structure of their inputs. Given the simplicity of our model, it is not to be expected that the exact dynamics of *Equation (4)* are replicated in real cells. However, the benefit of this simplicity is that we can now trace this behavior to its key ingredients, which we expect to be more general than the model itself: an excess of regulators, nonlinearity, and self-activation. In this section, we examine their role: first in our model (*Figure 5*), and then in more realistic circuits, relaxing our simplifying assumptions (*Figure 6*).

In *Figure 5*, the parameter $d$ on the horizontal axis is the strength of nonlinearity (see *Figure 2E*), from perfectly linear at $d = -\infty$, to strongly nonlinear at $d = 0$. The vertical axis corresponds to an increasing number of regulators $N_a$, which we label as in *Figure 2D*; for completeness, we also

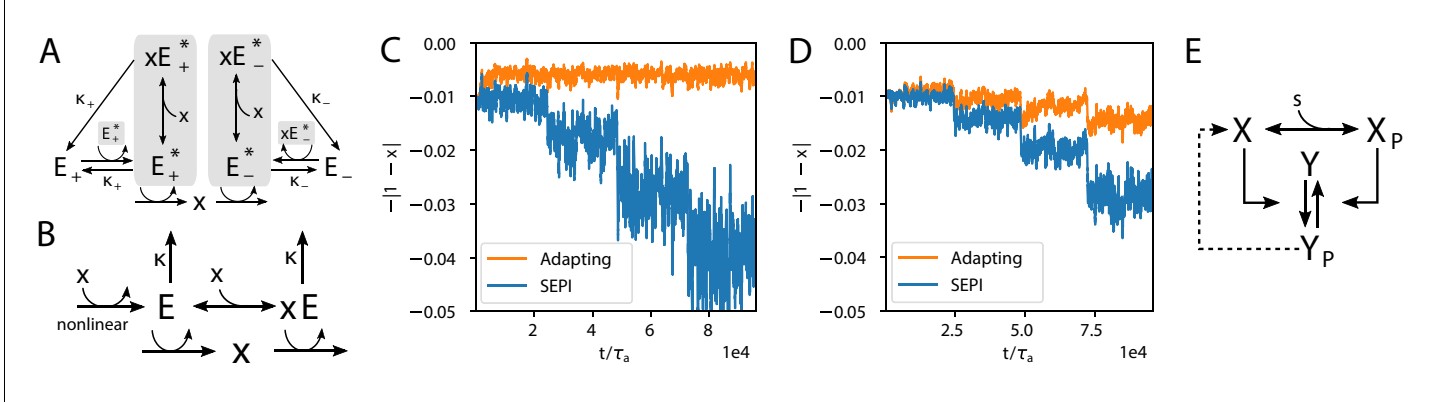

**Figure 6.** Realistic implementations. (**A**) An alternative circuit that implements the logic described above, but with different forms of the key ingredients, including a Hill-function nonlinearity. Here, the circuit is based on a pair of self-activating enzymes which can be in an active ($E^*$) or inactive state ($E$). For details see *Equation (5)*. (**B**) Another circuit capable of learning fluctuation variance to better maintain homeostasis of a quantity $x$. Synthesis and degradation of $x$ are catalyzed by the same bifunctional enzyme, whose production is regulated nonlinearly by $x$ itself. For more details see *Equations (6) and (7)*. (**C**) The circuit in panel A performs well at the homeostasis task of maintaining $x$'s concentration at 1, despite the changing variance of the input. For comparison, we've included a SEPI analogue of the circuit, described in Appendix 1, section 'Realistic biochemical implementations'. (**D**) Same as panel C, but with the circuit from panel B. Note that the 'SEPI' line is different here, and is now a SEPI analogue of the circuit in panel B. (**E**) Solid arrows: a common two-component architecture of bacterial sensory systems with a bifunctional histidine kinase (X) and its cognate response regulator (Y). Adding an extra regulatory link (nonlinear auto-amplification, dashed arrow) can endow this system with self-tuned reactivity learning the statistics of the input; see text.

include the simplest system with a single regulator co-activating both $x_1$ and $x_2$ (bottom row). Panel A examines the performance of our system as defined in *Equation (4)*, that is, with self-activation included. In panel B, we remove self-activation by omitting the prefactor $a_\mu$ in front of the max function in *Equation (4)*. The color scheme is chosen so that red indicates an improvement, and blue a deterioration, of the circuit performance relative to the SEPI architecture, which approximately corresponds to the point highlighted in *Figure 5B*. The difference between the labeled point and the SEPI architecture is that all models represented in *Figure 5* include a small degradation term, which becomes important in the nonlinear regime. For the SEPI-like case, its effect on performance is negligible (see Appendix 1, section 'Performance penalty from the degradation term') . Performance is averaged over five angles α; see Appendix 1, section 'Parameters used in figures'.

Unsurprisingly, the performance of the simple SEPI-like architecture can be improved by adding extra regulators (pink region in *Figure 5B*): each new regulator allows the system to respond more quickly in a yet another direction of perturbation, with which it is 'aligned'. However, such a strategy would have limited utility in a biological setting, since the marginal improvement per regulator must offset the cost of added complexity. The mechanism described here corresponds to the red area in *Figure 5A*. Importantly, in the presence of both nonlinearity and self-activation, even a single extra regulator ($N_a = 3$) can already provide a significant benefit.

*Figure 5A* shows that in the context of our model, the reported behavior requires $N_a$ to exceed $N_x$, and $d$ to be sufficiently large. However, these ingredients are more general than the specific implementation in *Equation (4)*. In our model, additional regulators were required because they supplied the slow degrees of freedom to serve as memory; such degrees of freedom could be implemented in other ways, for example, as phosphorylation or methylation (*Barkai and Leibler, 1997*). Similarly, while nonlinearity is essential (linear dynamics cannot couple to higher-order terms, such as fluctuation magnitude), its exact functional form may be changed while retaining the learning behavior (see Appendix 1, section 'Nonlinearity acts as a sensor of fluctuation variance'). Finally, the explicitly self-catalytic behavior of $a_\mu$ in our model is only one possible strategy for translating the stored memory into a faster response.

To demonstrate the generality of these ingredients, we constructed two circuits with very different architectures (*Figure 6A,B*), both reproducing the results of *Figure 3C–F*. These are not the only ways that the logic described above can be implemented; rather, our intention is to show that as

long as we keep the key elements, we can relax our simplifying assumptions, such as the form of the nonlinearity and self-activation, while still retaining the ability to learn.

The first of these proposed circuits (*Figure 6A*) is based on a pair of allosteric enzymes with the toy nonlinearity of *Figure 2E* replaced by more realistic cooperative binding, and implements dynamics otherwise very similar to those shown above. In this circuit, the enzymes $E_+$ and $E_-$ can be in an active or inactive state: The active form of $E_+$, which we denote $E_+^*$, catalyzes the production of $x$; similarly, $E_-^*$ catalyzes degradation of $x$. In addition, the active enzymes can bind to molecules of the metabolite $x$ to control the self-catalytic activity. The total concentration of $E_+^*$, bound and unbound, then plays the role of the activating regulator $a_+$ from above ($a_+ = [E_+^*] + [xE_+^*]$), while $E_-^*$ plays the role of the inhibitor $a_-$ ($a_- = [E_-^*] + [xE_-^*]$). The equations defining the dynamics are then:

$$\begin{cases} \tau_x \dot{x} = & \gamma_+ a_+ - \gamma_- x a_- - x D(t), \\ \tau_a \dot{a}_+ = & a_+ \dfrac{c_+^n}{c_+^n + x^n} - a_+ \kappa_+, \\ \tau_a \dot{a}_- = & a_- \dfrac{x^m}{c_-^m + x^m} - a_- \kappa_-. \end{cases} \tag{5}$$

Despite the extensive changes relative to *Figure 3A*, the system is still able to learn. *Figure 6C* compares its performance to a non-learning version with only the activating branch $a_+$, which is analogous to the single-activator SEPI architecture (compare to *Figure 3F*). For a detailed discussion of this more biologically plausible model, see Appendix 1, section 'A pair of allosteric enzymes'.

Our other proposed circuit (*Figure 6B*) differs significantly. Here, instead of seeking to match $P$ to $D$, the system maintains the homeostasis of a concentration $x$ perturbed by external factors. In this implementation, the production and degradation of $x$ are both catalyzed by a single bifunctional enzyme; the responsiveness of this circuit scales with the overall expression of the enzyme $E$, and larger fluctuations of $x$ lead to upregulation of $E$ due to the nonlinearity, as before. (For a detailed discussion, see e Appendix 1, section 'An architecture based on a bifunctional enzyme'.) Defining $A = a_+ + a_- = [E] + [xE]$ as the total concentration of the enzyme $E$ in both its bound and unbound states, the bound and unbound fractions are described by Hill equations,

$$a_+ = A \frac{c^m}{x^m + c^m}, \quad a_- = A - a_+. \tag{6}$$

The dynamics of our system are:

$$\begin{cases} \tau_x \dot{x} = & P_0 + \gamma_+ a_+ - \gamma_- x a_- - x D(t) \\ \tau_A \dot{A} = & -A\kappa + f(x). \end{cases} \tag{7}$$

Despite its compactness, this circuit is also able to learn (*Figure 6D*; compare to *Figures 3F* and *6C*).

Interestingly, this particular logic is very similar to a small modification of the standard two-component signaling architecture (*Figure 6E*). In this architecture, the signal $s$ determines the concentration of the phosphorylated form $Y_P$ of the response regulator $Y$; the rapidity of the response is determined by the expression of the histidine kinase $X$, present at a much lower copy number. Although the signaling architecture of *Figure 6E*, at least in some parameter regimes, is known to be robust to the overall concentrations of $X$ and $Y$(*Batchelor and Goulian, 2003*), this robustness property applies only to the steady-state mapping from $s$ to $Y_P$, not the kinetics. Thus, much like in *Figure 6B,a* nonlinear activation of $X$ by $Y_P$ (known as autoregulation [*Goulian, 2010*] or autoamplification [*Hoffer et al., 2001*], and shown as a dashed arrow in *Figure 6E*) would endow this signaling system with self-tuned reactivity that learns the statistics of the input.

## Discussion

In this paper, we have studied a regulatory architecture which is able to infer higher-order statistics from fluctuating environments and use this information to inform behavior. For concreteness, we phrased the regulatory task as seeking to match the production $\vec{P}$ of one or two metabolites to a rapidly fluctuating demand $\vec{D}$. Alternatively, and perhaps more generally, the circuits we constructed can be seen as maintaining the homeostasis in a quantity $\vec{x}$ that is continually perturbed by external

factors. We demonstrated that a simple architecture was capable of learning the statistics of fluctuations of its inputs and successfully using this information to optimize its performance. We considered one-dimensional and two-dimensional examples of such behavior.

In one dimension, learning the statistics of the input meant our circuit exhibited a self-tuned reactivity, learning to become more responsive during periods of larger fluctuations. Importantly, we have shown that this behavior can be achieved by circuits that are highly similar to known motifs, such as feedback inhibition (*Figure 2A–C*) or two-component signaling (*Figure 6B,E*). The latter connection is especially interesting: There are at least a few examples of two-component systems where autoamplification, a necessary ingredient for the learning behavior discussed here, has been reported (*Shin et al., 2006*; *Williams and Cotter, 2007*). Moreover, in the case of the PhoR/PhoB two-component system in *E. coli*, such autoamplification has been experimentally observed to allow cells to retain memory of a previously experienced signal (phosphate limitation; *Hoffer et al., 2001*), a behavior the authors described as learning-like. As reported, this behavior constitutes a response to the signal mean and is similar to other examples of simple phenotypic memory (e.g. *Lambert et al., 2014*); however, our analysis demonstrates that a similar architecture may also be able to learn more complex features. Such a capability would be most useful in contexts where the timescale of sensing could plausibly be the performance bottleneck. Since transcriptional processes are generally slower than the two-component kinetics, we expect our discussion to be more relevant for two-component systems with non-transcriptional readout, such as those involved in chemotaxis or efflux pump regulation.

In the two-dimensional case, our simple circuit was able to learn and unlearn transient correlation structures of its inputs, storing this information in expression levels of different regulators. Our argument was a proof of principle that, for example, the gut bacteria could have the means to not only sense, but also predict nutrient availability based on correlations learned from the past, including correlations that change over faster-than-evolutionary timescales, such as the life cycle (or dietary preferences) of the host. Importantly, we showed that this ability could come cheaply, requiring only a few ingredients beyond simple end-product inhibition.

The mechanism described here could suggest new hypotheses for the functional role of systems with an excess of regulators, as well as new hypotheses for bacterial function in environments with changing structure.

## Materials and methods

All simulations performed in Python 3.7.4. Simulation scripts reproducing all figures are included as *Source code 1*.

## Acknowledgements

We thank M Goulian, A Murugan, and B Weiner for helpful discussions. SL was supported by the German Science Foundation under project EN 278/10–1; CMH was supported by the National Science Foundation, through the Center for the Physics of Biological Function (PHY-1734030) and the Graduate Research Fellowship Program.

## Additional information

### Funding

| Funder | Grant reference number | Author |
| --- | --- | --- |
| Deutsche Forschungsgemeinschaft | EN 278/10-1 | Stefan Landmann |
| National Science Foundation | PHY-1734030 | Caroline M Holmes |
| National Science Foundation | Graduate Research Fellowship | Caroline M Holmes |

The funders had no role in study design, data collection and interpretation, or the decision to submit the work for publication.

### Author contributions
Stefan Landmann, Conceptualization, Software, Formal analysis, Investigation, Visualization, Writing - original draft, Writing - review and editing; Caroline M Holmes, Conceptualization, Investigation, Writing - original draft, Writing - review and editing; Mikhail Tikhonov, Conceptualization, Formal analysis, Supervision, Investigation, Writing - original draft, Writing - review and editing

### Author ORCIDs
Caroline M Holmes ⬤ https://orcid.org/0000-0001-9885-4933
Mikhail Tikhonov ⬤ https://orcid.org/0000-0002-9558-1121

### Decision letter and Author response
Decision letter https://doi.org/10.7554/eLife.67455.sa1
Author response https://doi.org/10.7554/eLife.67455.sa2

---

# Additional files
### Supplementary files
- Source code 1. Python 3.7.4 simulation code and scripts to reproduce all figures.

- Transparent reporting form

### Data availability
Python simulation scripts reproducing all figures in the paper from scratch, as well as pre-computed simulation data files for faster plotting, are provided as Source code 1 (and are also publicly available at https://doi.org/10.17632/5xngwv5kps.2).

---

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

## Appendix 1

This text contains supplementary information on modeling details, the analytical calculations, and the exact parameters used in figures.

### S1 Simple end-product inhibition

As the starting point of our regulatory architecture, we consider a basic form of end-product inhibition (SEPI). The environment is modeled by a time-dependent (externally induced) demand $D(t)$ for metabolite $x$ which is produced at a rate $P$ (controlled by the system); both $D$ and $P$ are defined in units of metabolite concentration per unit time. The depletion of the metabolite depends on its concentration $x$ and the demand $D$. Assuming first-order kinetics (or, alternatively, expanding a general function to linear order in small fluctuations of $x$) the dynamics of $x$ is:

$$\dot{x} = P - D\frac{x}{x_0}. \tag{S1}$$

Further, we consider the temporal evolution of the regulator activity $a$

$$\dot{a} = h(x,a). \tag{S2}$$

By linearizing $h(x,a)$ around the stationary values $(x_0, a_0)$ we get

$$\dot{a} = \lambda_x(x_0 - x) + \lambda_a(a_0 - a). \tag{S3}$$

To examine this equation, we introduce the Fourier transforms $\tilde{a}(\omega) = \mathcal{F}[a(t) - a_0]$ and $\tilde{x}(\omega) = \mathcal{F}[x(t) - x_0]$ and get

$$i\omega\tilde{a} = -\lambda_x\tilde{x} - \lambda_a\tilde{a} \quad \Rightarrow \quad \tilde{a}(\omega) = -\frac{\lambda_x\tilde{x}(\omega)(\lambda_a - i\omega)}{\lambda_a^2 + \omega^2}. \tag{S4}$$

*Equation (S4)* shows that if the support of $\tilde{x}(\omega)$ is restricted to high frequencies, $\omega \gg \lambda_a$, then the degradation term $\lambda_a(a_0 - a)$ in *Equation (S3)* is negligible. Including it would only add a restoring force, reducing the amplitude of fluctuations of $a$, and decreasing the performance of the system. Since we are interested in the regime of fast fluctuations of $x$, we choose to omit this term in the SEPI system. With $\lambda_x = 1/\lambda$ we arrive at the dynamics used in the main text:

$$\begin{cases} \dot{x} &= P - D\dfrac{x}{x_0} & \text{source-sink dynamics of metabolite } x \\ P &= aP_0 & \text{definition of regulator activity } a \\ \dot{a} &= \dfrac{x_0 - x}{\lambda} & \text{regulator activity inhibited by } x \end{cases}$$

In the nonlinear system (*Equation (3)* of the main text), however, fast fluctuations of $x$ can cause the growth of $a$ (as discussed in the section 'The nonlinearity as a sensor of the fluctuation variance'), thereby inducing slow frequency modes to its dynamics. Thus, in the nonlinear case, we cannot omit the degradation term.

### S2 Performance penalty from the degradation term

The model considered in the main text modifies the SEPI architecture as follows:

$$x_i = P_i/D_i \tag{S5}$$

$$P_i = \Sigma_\mu \sigma_{\mu i} a_\mu \tag{S6}$$

$$\tau_a \dot{a}_\mu = a_\mu \max\left(d, \sum_i \sigma_{\mu i}(x_0 - x_i)\right) - \kappa a_\mu. \tag{S7}$$

Consider the case of a static input. We observe that if $x_0$ is set to 1, as in the main text, the presence of the degradation term causes the equilibrium point of these dynamics to be displaced away

from $x_i = 1$. Therefore, for a static input, the performance of this system—the degree of mismatch between $P_i$ and $D_i$, or, equivalently, the deviation of $x_i$ from 1—is actually worse than the performance of the original SEPI.

While the case of a static input is irrelevant for the discussion in the main text, this slight offset leads to a performance penalty also for a fluctuating input. Indeed, time-averaging *Equation (S7)* shows that for any active regulator, we must have

$$\left\langle f\left(\sum_i \sigma_{\mu i}(1-x_i)\right)\right\rangle = \kappa,$$ (S8)

where $f$ is the nonlinearity in our equation, $f(\gamma) = \max(d, \gamma)$. Clearly, we will in general again have $\langle x_i \rangle \neq 1$; this is particularly obvious in the limit of small fluctuations when the nonlinearity is 'not active', such that $f(\gamma) = \gamma$.

This effect could be corrected by shifting $x_0$. In the interest of keeping our model as close to SEPI as possible, we chose not to do so: this penalty is only significant in the regime where no learning occurs, and is otherwise outweighed by the performance increase due to self-tuned responsiveness, with the additional benefit of simplifying the discussion in the main text. Even if optimizing $x_0$ could make the performance slightly closer to the optimal bound, this kind of fine-tuning seems irrelevant in a biological context.

## S3 Defining the system's responsiveness

In the main text, we use a measure for the 'responsiveness' of our system to changes in the demand $D(t)$. In this section it is shown in detail how this measure is defined. The central aim of the considered regulatory model is to match the time-dependent demand $\vec{D}$ with the regulated production $\vec{P}$. The temporal evolution of $\vec{P}$ is given by:

$$\dot{P}_i = \sum_\mu \sigma_{i\mu}\dot{a}_\mu,$$ (S9)

with

$$\tau_a \dot{a}_\mu = a_\mu \max\left(d, \sum_i \sigma_{\mu i}(1-P_i/D_i)\right) - \kappa a_\mu.$$ (S10)

For a static demand $D_i = 1$ the production relaxes to a constant value $P_i \approx 1$ (where we assumed small $\kappa$) and consequently $\dot{P}_i = 0$. A deviation $\delta D_i$ from the static demand will lead to a change of the production $P_i$ - the larger $\dot{P}_i$, the faster the response to the changed demand. Therefore, we define the responsiveness of the production $P_i$ to the demand $D_j$ as $\mathcal{R}_{ij} = \frac{d\dot{P}_i}{dD_j}$. When assuming small fluctuations the explicit expression for the responsiveness is then given by:

$$\frac{d\dot{P}_i}{dD_j} = \sum_\mu \sigma_{i\mu}\frac{d\dot{a}_\mu}{dD_j} \approx \sum_\mu \sigma_{i\mu}a_\mu \sigma_{\mu j}\frac{P_i}{D_j^2} \approx \sum_\mu \sigma_{i\mu}a_\mu \sigma_{\mu j}.$$ (S11)

As an example we consider the one-dimensional system studied in *Figure 3A-F* in the main text for which the responsiveness is

$$\mathcal{R} = \frac{d\dot{P}}{dD} = \sigma_{1+}a_+\sigma_{+1} + \sigma_{1-}a_-\sigma_{-1} = a_+ + a_-,$$ (S12)

where we used that $\sigma_{1+} = 1$ and $\sigma_{1-} = -1$.

## S4 Control theory calculation

The calculation below follows the standard procedure known as Linear-Quadratic Optimal Control (LQC). For more details, we refer the reader to standard sources (e.g. *Liberzon, 2011*).

The problem we set is minimizing $\langle(\vec{P} - \vec{D})^2\rangle$, where $\vec{D}$ follows the dynamics of *Equation (1)* in the main text,

$$\vec{D}(t + \Delta t) = \vec{D}(t) + M \cdot \left(\vec{\overline{D}} - \vec{D}(t)\right)\Delta t + \sqrt{2\Gamma\Delta t}\,\vec{\eta}. \tag{S13}$$

We then wish to calculate the optimal way of steering $\vec{P}$. For simplicity, we will set the mean $\overline{D} = 0$ (in the context of this abstract problem, this is equivalent to working with mean-shifted variables $\delta\vec{D} \equiv \vec{D} - \vec{\overline{D}}$ and similarly $\delta\vec{P} \equiv \vec{P} - \vec{\overline{D}}$). We can start by discretizing the above equation,

$$D_{t+1} = D_t - \tilde{M}D_t + \xi_t, \tag{S14}$$

where $\tilde{M} \equiv M\Delta t$, and $\xi$ has variance $2\Gamma\Delta t$. We seek to determine the optimal way to steer $P$; in other words, the function $\phi_t(P_t, D_t)$ ('control policy') dictating how $P$ should be changed in a given time step:

$$P_{t+1} = P_t + u_t \tag{S15}$$

$$u_t = \phi_t(P_t, D_t). \tag{S16}$$

We then can define our cost function, which combines a cost for the magnitude of $u_t$ (how quickly we can change $\vec{P}$), and the difference between $\vec{P}$ and $\vec{D}$:

$$\text{Cost} = \mathbb{E}\left(\rho\sum_{\tau=0}^{N-1}\|u_\tau\|^2 + \sum_{\tau=0}^{N}\|P_\tau - D_\tau\|^2\right). \tag{S17}$$

The $\phi(P_t, D_t)$ describing the optimal behavior is the one that minimizes this cost. In order to solve for $\phi(P_t, D_t)$, we follow standard control theory techniques and define the 'cost-to-go' function,

$$V_t(p_t, d_t) = \min_{\phi_t \ldots \phi_{N-1}} \mathbb{E}\left[\left(\rho\sum_{\tau=t}^{N-1}\|u_\tau\|^2 + \sum_{\tau=t}^{N}\|P_\tau - D_\tau\|^2\right)\middle| \begin{array}{l} P_t = p_t, D_t = d_t \\ D_{\tau+1} = (\mathbb{1} - \tilde{M})D_\tau + \xi_\tau \\ P_{\tau+1} = P_\tau + u_\tau \\ u_\tau = \phi_\tau(P_\tau, D_\tau) \end{array}\right]. \tag{S18}$$

This function defines the smallest cost of all remaining steps; in particular, the total cost that we are trying to minimize is $V_0(0,0)$. The cost-to-go satisfies the boundary condition

$$V_N(p,d) = \|p - d\|^2 \tag{S19}$$

and the following recursive relation:

$$V_t(p,d) = (p-d)^2 + \min_v\left\{\rho\|v\|^2 + \mathbb{E}_\xi V_{t+1}(p+v, (1-\tilde{M})d+\xi)\right\}. \tag{S20}$$

Since our system is Gaussian, this recursive relation can be solved by adopting a quadratic ansatz:

$$V_t(p,d) = p^\top A_t p - 2d^\top B_t p + d^\top C_t d + Q_t, \tag{S21}$$

Solving for the matrices $A_t$, $B_t$, $C_t$, and $Q_t$, gives us the following recursive relations:

$$\begin{cases} Q_t &= Q_{t+1} + 2\Gamma\Delta t\,\text{tr}\,C_{t+1} \\ A_t &= \mathbb{1} + \rho A_{t+1}(\rho + A_{t+1})^{-1} \\ B_t &= \mathbb{1} + \rho(\mathbb{1} - \tilde{M})B_{t+1}(\rho + A_{t+1})^{-1} \\ C_t &= \mathbb{1} + (\mathbb{1} - \tilde{M})\left[C_{t+1} - B_{t+1}(\rho + A_{t+1})^{-1}B_{t+1}^\top\right](\mathbb{1} - \tilde{M}) \end{cases} \tag{S22}$$

Since our problem is to minimize the cost at steady state (known in control theory as an 'infinite horizon' problem, $N \mapsto \infty$), we are interested in the fixed point of this mapping, specifically the matrices $A_{-\infty}$, $B_{-\infty}$ to which this mapping converges when we start from $A_N = B_N = C_N = \mathbb{1}$ and $Q_N = 0$ (as defined by **Equation (S19)**).

Since $A_N$ is the identity matrix, all $A_t$ are proportional to the identity matrix as well: $A_t = \alpha_t \mathbb{1}$, where $\alpha_t = 1 + \frac{\rho \alpha_{t+1}}{\rho + \alpha_{t+1}}$. The fixed point of this mapping is $\alpha = \frac{1 + \sqrt{1 + 4\rho}}{2} \geq 1$. Similarly, the fixed point of $B_t$ is $B = [\mathbb{1} - \frac{\rho}{\rho + \alpha}(\mathbb{1} - \tilde{M})]^{-1}$. Expressing this in terms of $\alpha$ only:

$$B = \alpha [\mathbb{1} + (\alpha - 1)\tilde{M}]^{-1}$$

With these expressions, the optimal 'control policy' is defined by the value of $v$ that minimizes *Equation S20*. This defines the optimal way to change $\vec{P}$ for a given observed $\vec{D}$:

$$u = \frac{1}{\alpha}\left([\mathbb{1} + (\alpha - 1)\tilde{M}]^{-1}(\mathbb{1} - \tilde{M})\vec{D} - \vec{P}\right), \tag{S23}$$

or, restoring the notations of the main text, including a non-zero $\overline{D}$:

$$u = \frac{1}{\alpha}\left([\mathbb{1} + (\alpha - 1)M\Delta t]^{-1}(\mathbb{1} - M\Delta t)(\vec{D} - \vec{\overline{D}}) - \vec{P}\right), \tag{S24}$$

This $u$ is the *exact* solution to the discrete version of the problem we considered. Since our simulations in this work use a discrete timestep, this is the form we use. Nevertheless, it is instructive to consider the small-$\Delta t$, large-CIP limit such that $\Delta t$ and $(\alpha - 1)\Delta t$ are both small compared to inverse eigenvalues of $M$. In this case we have, to first order in $\Delta t$:

$$u = \frac{1}{\alpha}\left([\mathbb{1} - \alpha M\Delta t](\vec{D} - \vec{\overline{D}}) - \vec{P}\right).$$

This leads to the following, and very intuitive, form of the optimal control dynamics:

$$\begin{aligned} \vec{D} &\mapsto \vec{D} - M\Delta t(\vec{D} - \vec{\overline{D}}) + \xi, \\ \vec{P} &\mapsto \vec{P} - M\Delta t(\vec{D} - \vec{\overline{D}}) + \frac{1}{\alpha}((\vec{D} - \vec{\overline{D}}) - \vec{P}). \end{aligned} \tag{S25}$$

In other words, at every step the change in $\vec{P}$ mirrors the average expected change in $\vec{D}$, with an extra term seeking to reduce their deviation. Note also that setting $\alpha = 1$ (infinite CIP) corresponds to steering $\vec{P}$ directly to the expected value of $\vec{D}$ at the next timestep, as expected.

## S5 The system makes use of correlations in the input

*Figure 4B* in the main text demonstrated that, as the fluctuating inputs become increasingly correlated, our architecture is able to outperform SEPI by an increasingly wide margin. The natural interpretation of this result is that the system is able to learn and exploit this correlation. Technically, however, one might note that this observation alone does not yet prove that our architecture is able to appropriately use the information it learned about specifically *correlation*. For example, it could be that strongly correlated inputs are somehow inducing a stronger increase in reactivity, causing the system to be generally faster, but without benefiting specifically from the correlated nature of its inputs.

Rather than focusing on excluding this specific scenario (which could be done by comparing the CIP values along the curves shown in *Figure 4B*), we will show that with a sufficient number of regulators, our architecture can perform better than the theoretical ceiling achievable by *any* strategy that assumes the inputs to be independent. This will formally prove that, at least for some parameters, our system's improved performance must necessarily make use of the correlation of its inputs. Although the argument is somewhat academic in nature (we will prove our point using $N_a = 25$ regulators), it is theoretically pleasing, and so we present it here.

Specifically, we consider a system subjected to inputs structured as shown in *Figure 3H*, with angle $\alpha = \pi/4$ so that the two inputs have the same variance. *Appendix 1—figure 1* shows the performance of our architecture for several values of the number of regulators $N_a$, plotted as curves parameterized by the degradation rate $\kappa$. The degradation rate controls how large the $a_\mu$ can become: a high value of $\kappa$ leads to lower average steady-state values of the regulator activities, causing the system to be less responsive to changes in $D$. Thus, $\kappa$ can be used to set the CIP of the

regulatory system, allowing us to plot these performance curves in the 'performance vs. CIP' axes traditional for control theory.

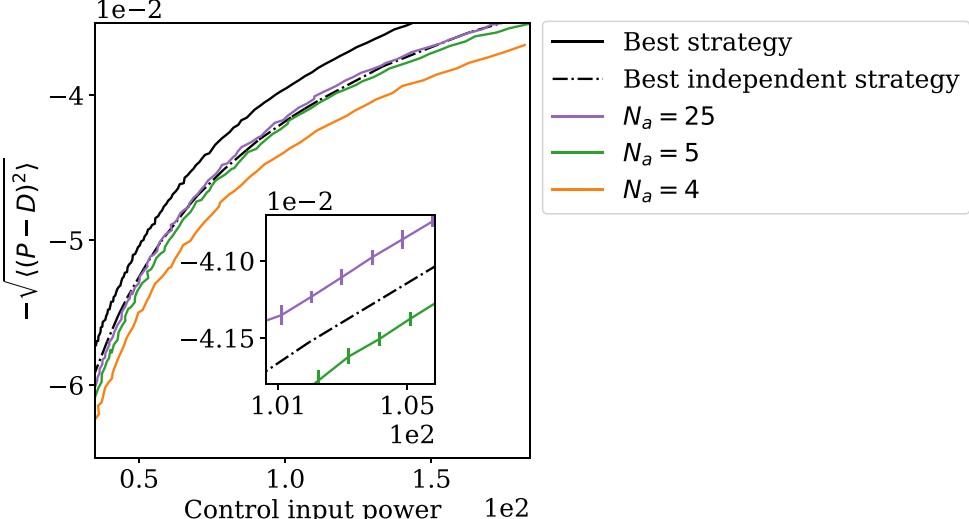

**Appendix 1—figure 1.** The adapting system can perform better than the best independence-assuming strategy.

Reassuringly, all these performance curves are located below the optimal control-theory ceiling computed for the true correlation structure of the input. However, the plot also shows the 'best independent' curve, defined as follows. Consider all possible matrices $M$ corresponding to uncorrelated inputs: $M = \begin{pmatrix} \lambda_1 & 0 \\ 0 & \lambda_2 \end{pmatrix}$. Each such $M$ defines a family of control strategies (that would have been optimal if this $M$ were the true $M$ governing the dynamics of the input); this family is indexed by a parameter $\rho$ as described above. A system following an (independence-assuming) strategy $M = \begin{pmatrix} \lambda_1 & 0 \\ 0 & \lambda_2 \end{pmatrix}$ while faced with the actual (partially correlated) inputs will exhibit a certain performance $\mathcal{P}(\lambda_1, \lambda_2, \rho)$ and a certain CIP, which we denote $\mathrm{CIP}(\lambda_1, \lambda_2, \rho)$. With these notations, the 'best independent' curve is defined as

$$\mathcal{P}(\mathrm{CIP} = \chi) = \max_{\lambda_1, \lambda_2}\{\mathcal{P}(\lambda_1, \lambda_2, \rho) \text{ for } \rho \text{ such that } \mathrm{CIP}(\lambda_1, \lambda_2, \rho) = \chi\}$$

We note that the correlated-input CIP is different from the independent-input CIP that a given strategy would have incurred if faced by the input for which it is optimal. In particular, while the latter can be computed analytically, the former has to be evaluated in simulations. This makes the optimization procedure computationally costly; thankfully, the symmetry ensured by choosing $\alpha = \pi/4$ allows restricting the search to isotropic strategies $M = \begin{pmatrix} \lambda & 0 \\ 0 & \lambda \end{pmatrix}$, reducing the problem dimensionality from three parameters $\{\lambda_1, \lambda_2, \rho\}$ to more manageable two $\{\lambda, \rho\}$.

The result is shown in *Appendix 1—figure 1* as a dashed line. As highlighted in the inset, with enough regulators, our architecture is indeed able to outperform the theoretical ceiling of the best independence-assuming strategy. Although $N = 25$ regulators is of course a regime irrelevant for biological applications, the aim of this argument was to formally prove a theoretical point, namely that the system as constructed must necessarily be making use of the correlation in the input signal, at least for some values of the parameters; by construction, the 'best independent' curve is a high bar to clear.

## S6 Model predictive control

When framing the problem in the main text, we discussed it as consisting of two tasks, learning the fluctuation structure of the environment and 'applying that knowledge' (*Figure 1B*), and treated the two as conceptually separate. In particular, the calculation in section 'S4 Control theory calculation' is known as linear-quadratic Gaussian control (LQG) that assumes the correct model of the environment to already be known. This separation was done to simplify presentation, allowing us to encapsulate the control theory discussion, potentially less familiar to our audience, to a smaller portion of the narrative. In addition, the LQG calculation is simple, exact, and provides an upper bound on how well any other control strategy could perform.

Mechanistically, however, the two tasks are inseparable. In the language of control theory, our architecture implements an example of model predictive control: a strategy where the system response is governed by a continually updated internal model of the environment (here, the activity levels of the regulators, which encode the learned correlations between the inputs).

How could one distinguish a predictive vs. non-predictive control scheme in practice, when measuring all internal variables to decipher whether or not they encode an 'internal model of the environment' is infeasible? For our scheme, its 'predictive' ability manifests itself as follows. Imagine exposing our system to two inputs $\vec{D}(t) = \langle \vec{D} \rangle + (\delta D_1(t), \delta D_2(t))$ which for a period of time are strongly correlated, with $\delta D_1(t) \approx \delta D_2(t)$. The learning process will drive the responsiveness matrix from one that was initially isotropic to one aligned with the correlated direction (in the notation of the main text, $\alpha = \pi/4$). Compare now the response of the system to an increase in $D_1$: $(\delta D_1, \delta D_2) = (a, 0)$. The naïve (untrained) system would respond by increasing $P_1$ only, as would the SEPI architecture. In contrast, the 'trained' system, having learned the correlation between $D_1$ and $D_2$, responds by upregulating both $P_1$ and $P_2$ together. In this hypothetical example, our analysis predicts that deleting seemingly superfluous regulators would hinder or remove this ability (depending on the implementation, possibly without even affecting fitness in a static environment).

This is the behavior expected of a predictive controller: Under a model where $\delta D_1$ and $\delta D_2$ are strongly correlated, one expects this state of the environment to relax back to the diagonal $\delta D_1 = \delta D_2$ in the near future. This kind of associative learning is experimentally measurable and, on an evolutionary scale, is well-known. Our scheme provides a mechanism for implementing this behavior on a physiological timescale. Another mechanism focusing on binary associations was previously described by *Sorek et al., 2013*.

## S7 Nonlinearity acts as a sensor of fluctuation variance

In the main text, we argue that the nonlinearity in the dynamics of the regulator concentrations acts as a senor for the variance of the fluctuations. To see this, we consider the dynamics of one regulator that is controlling the production of one metabolite:

$$\tau_a \dot{a} = a \max(d, \, 1 - P/D) - \kappa a. \tag{S26}$$

To simplify notation, we define $\gamma := 1 - P/D$. Since the dynamics of $a$ are slow compared to $D$, the fluctuations of $\gamma$ are on a faster timescale than the regulator dynamics. If the fluctuations of $\gamma$ are small, the nonlinearity in the $\max$ function is 'not activated': $\max(d, \gamma) = \gamma$. In this case, the temporal average of $\max(d, \gamma)$ is zero. In contrast, if the fluctuations are strong enough, the nonlinearity is activated (see *Appendix 1—figure 2*). Then, the temporal average is positive, leading to an additional growth of $a$. Due to the resulting larger values of $a$, the response of the system becomes faster, making the match between $P$ and $D$ better and thus serving to decrease the variance of $\gamma$. As a result, the final average steady-state regulator concentration is reached if the system has decreased the variance of $\gamma$ sufficiently by increasing the rapidity of its response.

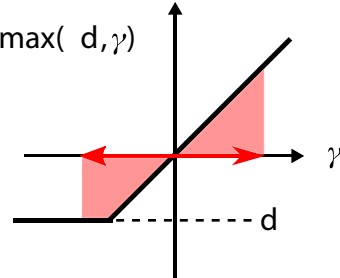

**Appendix 1—figure 2.** The nonlinearity in the regulatory architecture. If the fluctuations of the input $\gamma = \sum_i \sigma_{\mu i}(1 - x_i)$ are large enough, the average over the nonlinearity is positive, causing additional growth of the regulator concentration $a$.

This argument makes it clear why the specific choice of nonlinearity is not particularly important. If $\dot{a} = f(x)$, then the static steady state satisfies $f(x_0) = 0$. For a fast-fluctuating input this becomes

$$\dot{a} = \langle f(x) \rangle = f(x_0) + \frac{1}{2}\langle (x - x_0)^2 \rangle f''(x_0) + \ldots$$

For any nonlinear $f$, as long as $f(x_0) = 0$, the displacement of the original steady state will be determined by higher order statistics of the input. In particular, the rectified-linear nonlinearity in our equations can be replaced, for example, by a Hill function. Note that for the argument presented here, the eventual saturation of the response at large argument is irrelevant: the system will retain all the relevant behavior as long as the new nonlinearity is cup-convex in a sufficiently large vicinity of its static fixed point; see section 'S9.1 A pair of allosteric enzymes' for an explicit example.

The assumption $f(x_0) = 0$ is not innocuous; in general, of course, the value of $\langle f(x) \rangle$ is sensitive not only to the variance of $x$ (or other higher-order terms), but also to its mean, and building a system that is sensitive to specifically the variance requires adapting to the mean first. In our model, this is automatically accomplished by the underlying end-product inhibition architecture, which adapts the mean $P$ to mean $D = \overline{D}$, after which $x$ fluctuates around 1, no matter the value of $\overline{D}$.

## S8 The minimal $\Gamma$ needed to initiate adaptation

*Figure 4A* in the main text includes arrows indicating theoretically derived threshold values of $\Gamma$ above which our system (with a given $\sigma_{\mu i}$) will begin to adapt its timescale of response, deviating from SEPI in its behavior. Here, we show how this threshold value of $\Gamma$ can be determined.

As discussed in the main text, at static input ($\Gamma = 0$) only $N_x$ out of $N_a$ regulators can be active. Consider the regulators that remain inactive in the static case, and imagine gradually increasing the fluctuation magnitude. Recall the equation for regulator activity dynamics:

$$\tau_a \dot{a}_\mu = a_\mu \max\left(d, \sum_i \sigma_{\mu i}(1 - P_i/D_i)\right) - \kappa a_\mu. \tag{S27}$$

After introducing $\gamma_\mu = \sum_i \sigma_{\mu i}(1 - P_i/D_i)$ we can write

$$\tau_a \dot{a}_\mu = a_\mu \left(\max(d, \gamma_\mu) - \kappa_\mu\right) = a_\mu \Delta_\mu. \tag{S28}$$

If we chose $a_\mu$ as one of the regulators that remained inactive in the static case, we have $\Delta_\mu < 0$ at $\Gamma = 0$; as we begin increasing the fluctuation magnitude, the time-averaged $\Delta_\mu$ will at first remain negative. The threshold $\Gamma$ we seek is the one where the first (time-averaged) $\Delta_\mu$ crosses into positive values. It is clear that if the fluctuations of $\gamma_\mu$ are so small that $\max(d, \gamma_\mu) = \gamma_\mu$ at all times, the system does not adapt. On the other hand, if the fluctuations are large enough and fast compared to the response of the system, they generate an effective additional growth of $a_\mu$. To first order, this additional growth term is proportional to the standard deviation $\sqrt{\omega_\mu}$ of $\gamma_\mu$. Therefore, we expect the fluctuations to cause a growth of $a_\mu$ if the additional growth term is large compared to $\kappa$, i.e. $\sqrt{\omega} c \cdot \kappa$, with $c$ a constant of order 1.

The approximate value of $c$ can be determined using the following argument. With $d = 0$, and assuming that $\gamma_\mu$ is, first, fluctuating on a fast timescale compared to $\tau_a$ and, second, is Gaussian with mean $\overline{\gamma}_\mu$ and variance $\omega_\mu$, we can average over the fluctuations in *Equation (S28)*:

$$\langle \Delta_\mu \rangle = \frac{\overline{\gamma}_\mu}{2} + \sqrt{\frac{\omega_\mu}{2\pi}} \exp\left(-\frac{\overline{\gamma}_\mu^2}{2\omega_\mu}\right) + \frac{\overline{\gamma}_\mu}{2}\mathrm{erf}\left(\frac{\overline{\gamma}_\mu}{\sqrt{2\omega_\mu}}\right) - \kappa. \tag{S29}$$

The system is in a stable steady state if either $\langle \Delta_\mu \rangle = 0$ and $a_\mu \geq 0$ or $\langle \Delta_\mu \rangle < 0$ and $a_\mu = 0$. In the non-trivial first case we get the condition $\overline{\gamma}_\mu \leq \kappa$. Approximating $\overline{\gamma}_\mu \approx 0$ one sees that the average growth rate $\langle \Delta_\mu \rangle$ is positive if $\sqrt{\omega_\mu} > \sqrt{2\pi}\kappa$, so that $c = \sqrt{2\pi}$. If this condition is satisfied, $a_\mu$ continues its growth until the separation of timescales between $\gamma_\mu$ and $\tau_a$ becomes invalid and $\omega_\mu$ decreases; this is the mechanism by which the system adapts to fast fluctuations.

The variance $\omega_\mu$ can be derived from the statistical properties of $D$. If the fluctuations of the demand $D$ are small it holds that $\omega_\mu \approx \delta\hat{D}^T \vec{\sigma}_\mu \delta\hat{D}$ where $\delta\hat{D}$ is the covariance matrix of the stationary probability distribution of the fluctuations $\delta\vec{D}$ with $\langle \delta D_1^2 \rangle = \Gamma\left(\frac{\cos^2\alpha}{\lambda_1} + \frac{\sin^2\alpha}{\lambda_2}\right)$ and $\langle \delta D_1 \delta D_2 \rangle = \Gamma \cos\alpha \sin\alpha\left(\frac{\lambda_1 - \lambda_2}{\lambda_1\lambda_2}\right)$. The variance $\omega_\mu$ is then given by $\omega_\mu = \vec{\sigma}_\mu^T \delta\hat{D}\, \vec{\sigma}_\mu$ and the minimal value of $\Gamma$ is set by the largest $\omega_\mu$ of the considered system.

## S9 Realistic biochemical implementations

In the main text, we proposed a simple model of metabolic regulation which highlighted the necessary properties for learning environment statistics, namely an excess of regulators $a_\mu$, self-activation, and a nonlinear regulation of $a_\mu$ by the metabolite concentrations $x_i$. To show how these properties can enable more realistic systems to learn the statistics of a fluctuating environment, here we present two biochemical implementations. The first of these implements dynamics nearly identical to those described in the main text, and the second, illustrated in *Figure 5b*, bears resemblance to two-component systems. As described in the main text, we do not necessarily expect either of these networks to be implemented in real biological systems 'as is'. Instead, we use these to illustrate the diversity of systems that could use the logic described in this paper to learn statistics of their environment. For simplicity, we consider the one-dimensional case (first column of *Figure 3* in the main text).

### S9.1 A pair of allosteric enzymes

The circuit is shown in *Appendix 1—figure 3*. The enzymes $E_+$ and $E_-$ can be in an active or inactive state: The active form of $E_+$, which we denote $E_+^*$, catalyzes the production of $x$; similarly, $E_-^*$ catalyzes degradation of $x$. In addition, we postulate that active enzymes can bind to molecules of the metabolite $x$, which controls self-catalytic activity (see diagram). The total concentration of $E_+^*$, bound and unbound, plays the role of the activating regulator $a_+$ from the main text ($a_+ = [E_+^*] + [xE_+^*]$), while $E_-^*$ plays the role of the inhibitor $a_-$ ($a_- = [E_-^*] + [xE_-^*]$).

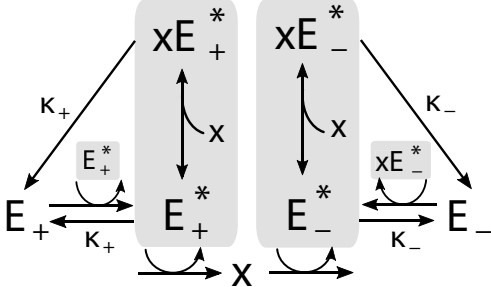

**Appendix 1—figure 3.** Implementation of the regulatory mechanism based on a pair of self-activating enzymes which can be in an active ($E^*$) or inactive state ($E$). Gray shading indicates catalysts of reactions.

The same regulatory structure could be realized with transcription factor regulation, with the role of the active enzymes ($E_+$ and $E_-$) played by two transcription factors. In this version, activation/deactivation of enzymes is replaced by the simpler process of transcription factor synthesis/degradation. For concreteness, here, we focus on the enzymatic case, largely because we expect architectures like ours to be more relevant in fast-responding circuits, which tend to be non-transcriptional. However, except for the difference of timescales, the dynamics of the two versions would otherwise be identical; in particular, both implementations would 'learn' in the same way.

For this discussion, it is convenient to have timescales of dynamics of $a_\mu$ and $x_i$ encoded as explicit parameters. Assuming first-order kinetics, the dynamics of the network can then be described by:

$$\begin{cases} \tau_x \dot{x} = & \gamma_+ a_+ - \gamma_- x a_- - x D(t), \\ \tau_a \dot{a}_+ = & a_+ \dfrac{c_+^n}{c_+^n + x^n} - a_+ \kappa_+, \\ \tau_a \dot{a}_- = & a_- \dfrac{x^m}{c_-^m + x^m} - a_- \kappa_-. \end{cases} \tag{S30}$$

Here, we assume that the metabolite $x$ is much more abundant than the active enzymes $E_+^*$ and $E_-^*$, meaning that the relative amount of bound $x$ is very small. This allows us to neglect, in the dynamics of $x$, the source and sink terms due to binding and unbinding of $x$ to the enzymes. We also assume that this binding and unbinding occurs on a much faster timescale than all other processes.

*Appendix 1—figure 4* shows an example of simulation results for these dynamics (for the full list of parameters used, see section 'S11 Parameters used in figures'). We see that the system reacts to an increasing variance of environmental fluctuations (A) by increasing regulator activities (B). The figure also shows the behavior of a SEPI system which only consists of one $a_+$ regulator described by the dynamics in *Equation (S30)*. *Appendix 1—figure 4C* shows that the response strength, defined as discussed in the SI section 'Defining the system's responsiveness,'

$$\mathcal{R} = \frac{d\dot{P}}{dD} \approx \gamma_+ a_+ \frac{n c_+^n}{\left(c_+^n + 1\right)^2} + \gamma_- a_- \frac{m c_-^m}{\left(c_-^m + 1\right)^2}, \tag{S31}$$

is increasing due to the changed regulator activities. Finally, *Appendix 1—figure 4D* compares the performance of the system *Equation (S30)* with the corresponding SEPI system (which, again, we define by the same equations as *Equation (S30)*, except without the $a_-$ regulator). Similar to *Figure 3F* in the main text, the performance of the adapting system does not change as the variance of the stimulus increases, while the SEPI system becomes worse.

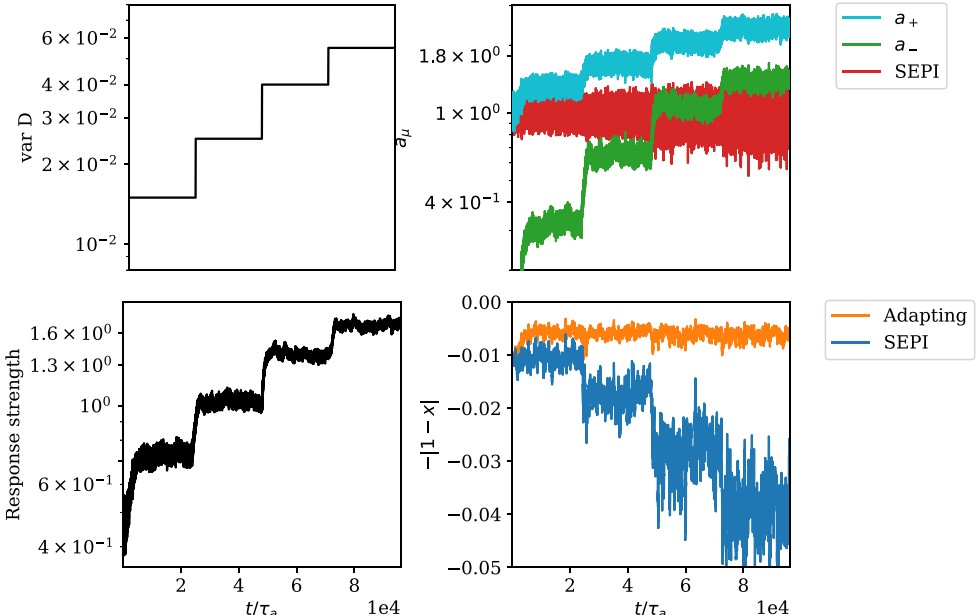

**Appendix 1—figure 4.** Adaptation of responsiveness to an increasing variance of environmental fluctuations. (**A**) Step-wise increase of the variance of $D$. (**B**) Time-series of regulator concentrations, where $a_+$ and $a_-$ correspond to the total concentrations of $t_+$ and $t_-$ respectively. (**C**) The responsiveness of the system as defined in *Equation (S31)*. (**D**) The deviation of the metabolite concentration $x$ from its target value.

## S9.2 An architecture based on a bifunctional enzyme

For the reader's convenience, we reproduce this circuit in *Appendix 1—figure 5* (identical to *Figure 6B* in the main text). As described in the main text, for greater generality, we will here rephrase the task: instead of matching production to demand, we will think of maintaining the homeostasis of a quantity $x$ perturbed by external factors. For example, instead of being a metabolite concentration, $x$ could be osmolarity mismatch, and our circuit a hypothetical architecture for active control of osmotic pressure. In this interpretation, the enzyme $E$ might be a mechanosensor triggered by tension in the membrane or cell wall, while 'production' and 'degradation' of $x$ could be activities of opposing pumps, or regulators of glycerol export or metabolism.

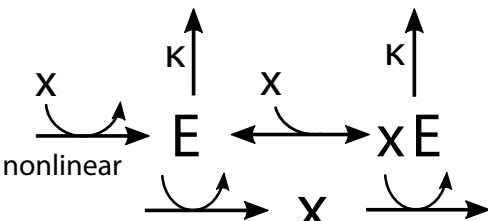

**Appendix 1—figure 5.** Regulation by allosteric forms of one enzyme $E$. The unbound form $E$ activates the production of $x$, while the bound form $xE$ promotes its degradation. The synthesis of $E$ is regulated nonlinearly by the metabolite concentration $x$.

To simplify our language when describing the terms in the specific equations we simulate, we will still talk of a metabolite $x$ being produced and degraded. However, to better accommodate alternative interpretations, the regulator activities will now be defined so that $a_+$ and $a_-$ would be equal on average (for example, the activities of pumps in opposite directions should, on average, balance).

This homeostasis-maintaining formulation is in contrast to *Figure 3D* in the main text, where regulators satisfied the constraint $\langle a_+ - a_- \rangle = \overline{D} = 1$.

The production and degradation of $x$ are catalyzed by a bifunctional enzyme that changes its activity when bound to $x$ forming the compound $xE$. The concentration of the unbound form $E$ corresponds to the activating regulator, $a_+ = [E]$, and increases the production $P$ of $x$, while $xE$ plays the role of the inhibiting regulator, $a_- = [xE]$, and promotes the degradation of $x$.

As above, we assume first-order kinetics for the production and degradation of $x$, and that the binding kinetics are fast compared to the other timescales in the problem. Defining $A = a_+ + a_- = [E] + [xE]$ as the total concentration of the enzyme $E$ in both its bound and unbound states, the bound and unbound fractions are described by Hill equations:

$$a_+ = A\frac{c^m}{x^m + c^m}, \quad a_- = A - a_+ = A\frac{x^m}{x^m + c^m}. \tag{S32}$$

These expressions make it explicit that a small change in the concentration $x$ induces a change in $a_+$ and $a_-$ that is proportional to their sum, $A = a_+ + a_-$. In this way, the circuit shown in *Appendix 1—figure 6* does include an element of self-activation (one of our 'key ingredients'), even though no interaction in the diagram is explicitly self-catalytic.

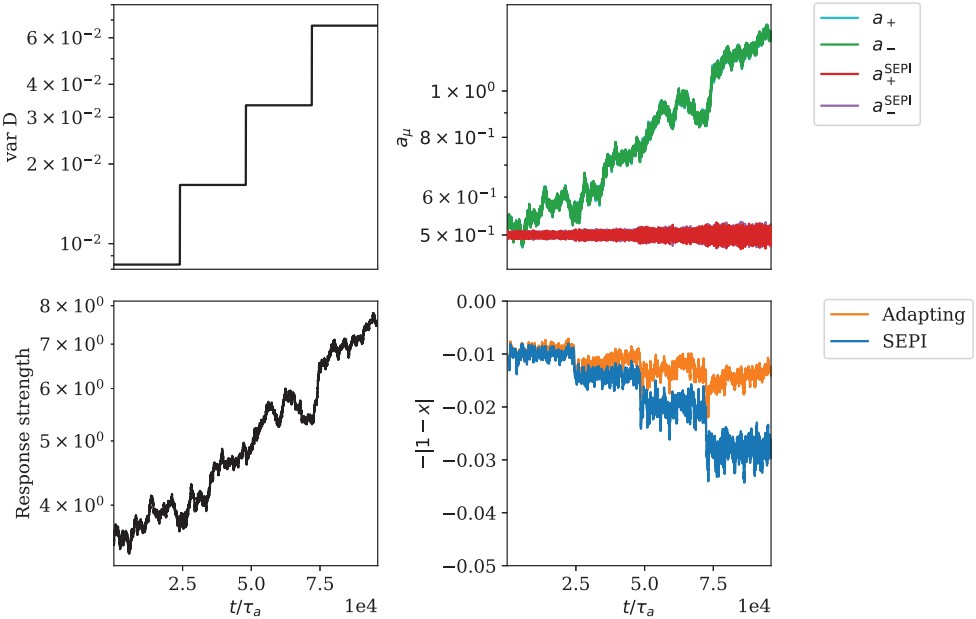

**Appendix 1—figure 6.** Adaptation of responsiveness for the circuit architecture based on a bifunctional enzyme. (**A**) The variance of $D$ is increased step-wise. (**B**) Change of regulator activities. The regulator activities $a_+$ and $a_-$ overlap strongly and cannot be distinguished in this panel. (**C**) The response strength of the system. (**D**) The mismatch of the metabolite concentration $x$ from its target value.

With these notations, the dynamics of our system are:

$$\begin{cases} \tau_x \dot{x} = P_0 + \gamma_+ a_+ - \gamma_- x a_- - xD(t) \\ \tau_A \dot{A} = -A\kappa + f(x), \end{cases} \tag{S33}$$

where we assumed that modifying the enzyme $E$ does not significantly affect the quantity $x$. (In the metabolic formulation, this corresponds to the assumption that $x$ is much more abundant than $E$, so that the sequestration of $x$ by $E$ has negligible effect on the free concentration of $x$; in the osmotic formulation, the act of triggering a mechanosensor has negligible instantaneous effect on pressure). In the second equation, the synthesis of the enzyme $E$ depends nonlinearly on the metabolite concentration $x$. The specific form of nonlinearity does not significantly affect the results, as long as it is sufficiently cup-convex in the vicinity of the operating point: As described in the section 'S7 Nonlinearity acts as a sensor of fluctuation variance', we can think of the nonlinearity $f(x)$ as a 'sensor' for

the variance of environmental fluctuations. Whenever fluctuations in $D(t)$ increase such that the current responsiveness of the system fails to maintain the homeostasis of $x$ within previous bounds of fluctuation magnitude, the fluctuations of $x$ will lead to growth of $A$, increasing the responsiveness until it is again able to reduce the fluctuations of $x$. In our simulations we chose a Hill function with cooperativity 4 (see section 'S11 Parameters used in figures').

*Appendix 1—figure 6* shows simulation results for this system. As in the first column of *Figure 3*, the variance of $D$ is increased and the response of the system to this change is monitored. We see that the regulator concentrations correspondingly increase, causing a larger response strength $|\frac{dx}{dx}| \approx 1 + \frac{2\gamma E c^m m}{(1+c^m)^2}$. The increase in response strength is able to compensate for most of the performance loss, which shows that the system successfully adapts its timescale of response. This is in contrast to the 'SEPI-like' system with a fixed value $A = 1$, which cannot adapt its responsiveness and whose performance drops with every increase in fluctuation variance.

## S10 Adaptation to static demand

In the main text, we argue that the production $\vec{P}$ of the proposed system *Equation (3)* adapts to any static demand $\vec{D}$. The full dynamics of the system is

$$\tau_a \dot{a}_\mu = a_\mu \max\left(d, \sum_i \sigma_{\mu i}(1 - P_i/D_i)\right) - \kappa a_\mu. \tag{S34}$$

With a static demand, *Equation (S34)* possesses the same fixed points as the simplified dynamics:

$$\tau_a \dot{a}_\mu = a_\mu \left(\sum_i \sigma_{\mu i}(1 - P_i/D_i) - \kappa\right). \tag{S35}$$

These dynamics have a Lyapunov-function

$$F(\{a_\mu\}) = -\sum_i \frac{1}{2D_i}(P_i - D_i)^2 - \kappa \sum_\mu a_\mu. \tag{S36}$$

This can be verified by considering the temporal change of $F$

$$\frac{dF}{dt} = \sum_\mu \frac{\partial F}{\partial a_\mu}\frac{da_\mu}{dt} = \sum_\mu a_\mu \Delta_\mu^2 > 0, \tag{S37}$$

with $\Delta_\mu = \sum_i \sigma_{\mu i}(1 - P_i/D_i) - \kappa$. Thus, $F$ always increases and is obviously bound from above. For small $\kappa$, the maximum of $F$ is reached for $\vec{P} \approx \vec{D}$, showing that the system adapts to any static demand.

## S11 Parameters used in figures

If not stated otherwise, we use the following parameters: $\overline{D} = 1$, $\Gamma = 0.35$, $d = 0$, $\kappa = 0.015$, $\tau_a = 3$, $\alpha = 45°$, $\lambda_1 = 8.75$, $\lambda_2 = 875$, $dt = 1/\lambda_2 \approx 1.14 \cdot 10^{-3}$. Since the demand $D_i$ is modeled by a stochastic process which is, in principle, unbounded, there is a non-zero probability that the demand $D_i$ becomes negative. To prevent this behavior in the simulations we set $D_i = 0.01$ if $D_i < 0.01$.

### Figure 3C–F

Fluctuations: In 1D the matrix $M$ only has one element which we set to $M = 7.5$, $\Gamma = [0.048, 0.082, 0.16, 0.3]$.
System: $\kappa = 0.03$.
Timescales: $\tau_a = \tau_a^{SEPI} = 1$.
*Figure 3F* shows a running average over 100 steps.

### Figure 3I–L

Fluctuations: $\alpha = [-60, 30, -30, 60]$
System: $N_a = 5$, $\tau_a = 1/\lambda_1$, $\kappa = 0.02$.
SEPI: For a fair comparison, the timescale of SEPI is chosen such that its responsiveness matches the faster responsiveness of the $N_a = 5$ adapting system (measured in an environment with an isotropic $M$ with the same determinant as used in *Figure 3J–L*): $\tau_a^{SEPI} = \tau_a/4.9$.
For visualization purposes, to prevent long transients after changes of the activation angle, the regulator activities were capped from below at $a_\mu = 0.1$.
*Figure 3L* shows a running average over 100 steps.

### Figure 4A

Fluctuations: $\Gamma$ from 0 to 0.1 in 40 equidistant steps.
Timescale SEPI: $\tau_a^{SEPI} = \tau_a = 3$.
Simulation length: $5 \cdot 10^7$ steps.

### Figure 4B

Fluctuations: $\Gamma = 0.05$, anisotropy A=[ 1, 1.05,1.1, 1.15, 1.2 , 1.25, 1.3 , 1.35, 1.4, 1.45, 1.5, 1.55, 1.6 , 1.65, 1.7, 1.75, 1.8, 1.85, 1.9, 1.95, 2, 2.1, 2.2, 2.3, 2.4, 2.5, 2.6, 2.7, 2.8, 2.9, 3.25, 3.5, 4, 4.5, 5, 6, 7, 8, 9, 10]. For each value of $A$ and $\lambda_2$ are chosen as: $\lambda_1 = \frac{1+A^2}{rA^2}$, $\lambda_2 = A^2\lambda_1$ with $r = 1/8.75 + 1/875$.
Timescale SEPI: $\tau_a^{SEPI} = \tau_a = 3$.
Simulation length: $5 \cdot 10^7$ steps.

### Figure 5A and B

Fluctuations: Results averaged over activation angles $\alpha = [45, 85, 125, 165, 205]$.
System: $\kappa = 0.02$, $\tau_a = 1/\lambda_1$.
Simulation length: $10^7$ steps.

### Figure 6C and Appendix 1—figure 4

The parameters in the simulation were chosen so as to ensure that, first, $\tau_x \ll \tau_a$; and second, the steady-state $x$ stays at 1 (this is analogous to setting $x_0 = 1$ in the main text). Specifically, we used: $\tau_x = 0.01$, $\tau_a = 1$, $\gamma_+ = \gamma_- = 1$, $n = 2$, $m = 2$, $c_+^n = 0.5$, $c_-^m = 2$, $\kappa_+ = 1.0025 \frac{1}{c_+^n + 1}$, $\kappa_- = 1.0025 \frac{c_-^m}{c_-^m + 1}$. The parameters describing the fluctuations of $D$ are chosen as: $\overline{D} = 1$, $M = 1$, $\Gamma = [0.015, 0.025, 0.04, 0.055]$.

A brief explanation: While the parameter list is long, there is a simple reasoning which sets most of these choices, and explains how the parameters of this model need to be related to each other in order for the adaptation of responsiveness to occur. First of all, we assume that the concentration $x$ changes on a much faster timescale than the regulator concentrations $a$; here we choose $\tau_a = 1$ and $\tau_x = 0.01$. Further, the average of $D(t)$ is chosen to be equal to one. Then, for small fluctuations of $D$ we have $x \approx \gamma(a_+ - a_-)$. On the other hand, the non-trivial fixed points of the regulator concentration dynamics are reached if

$$\frac{c_+^n}{c_+^n + x^n} = \kappa_+ \quad and \quad \frac{x^m}{c_-^m + x^m} = \kappa_-. \tag{S38}$$

Thus, we can set the equilibrium point of $x$ by choosing $\kappa_+$, $\kappa_-$, $c_+$ and $c_-$. Here, without loss of generality, we choose that the fixed point is reached at $x = x_0 = 1$ by setting

$$\frac{c_+^n}{c_+^n + 1} = \kappa_+ \quad and \quad \frac{1}{c_-^m + 1} = \kappa_-. \tag{S39}$$

For the sought-after effect to occur, the fast fluctuations of $x$ around $x_0 = 1$ need to result in an effective additional growth of $a_+$ and $a_-$, providing a 'sensor' for the variance of $D$. One possibility

to get this behavior is to set $c_-^m = 2$ and $c_+^n = 0.5$. To avoid the regulator concentrations to grow indefinitely, $\kappa_+$ and $\kappa_-$ need to be a little larger than the determined values in *Equation (S39)*. Finally, the parameter $\gamma$ can be chosen rather freely; here we choose $\gamma = 1$. Simulation length: $3.2 \cdot 10^7$ steps with $dt = 3 \cdot 10^{-3}$. Panels 6C and S4 D show a running average over 100 timesteps.

### *Figure 6D* and *Appendix 1—figure 6*:

We used the following parameters for the simulations: $\tau_x = 1$, $\tau_A = 25$, $P_0 = 1$, $\gamma_+ = \gamma_- = 5$, $\kappa = 10^{-3}$, $c^m = 1$, $m = 1$. The nonlinearity was chosen as: $f(x) = d \frac{x^n}{x^n + c_1^n} - \beta$ with $d = 10$, $c_1^n = 10$, $n = 4$, $\beta = 5 \cdot 10^{-4}$. The parameters describing the fluctuations of $D$ are set to: $M = 3$, $\Gamma = [0.025, 0.05, 0.1, 0.2]$. For the mechanism to work, the timescales need to fulfill $\tau_A \gg \tau_x$. The parameters $P_0$, $m$ and $c$ are set by the choice of the fixed-point $x_0$ (here $x_0 = 1$). Simulation length: $3.2 \cdot 10^7$ steps with $dt = 3 \cdot 10^{-3}$. Panels 6D and S6 D show a running average over 100 timesteps.

### Appendix 1—figure 1

System: $\kappa$ from 0.01 to 0.025 in steps of size $1.25 \cdot 10^{-4}$.
Simulation length = $5 \cdot 10^7$ steps.
For each simulation, the performance was averaged over the last $10^7$ time steps.

### Appendix 1—figure 1 inset

All system parameters as in Figure S1 except for: $\kappa$ from 0.013 to 0.014 in steps of size $2.5 \cdot 10^{-5}$.
Simulation length: $10^8$ steps.
For each simulation, the performance was averaged over the last $2 \cdot 10^7$ steps.
The results are binned in 20 equal-width bins.

