## [Decision Letter]

**Acceptance summary:**

This paper shows how a common biochemical motif enables an organism to learn parameters describing the statistical structure of environmental fluctuations. The inferred parameters (here, the mean and variance) can be used to mount an adaptive response. The work shows a simple design principle to achieve control over phenotype, homeostasis and growth.

**Decision letter after peer review:**

Thank you for submitting your article "A simple regulatory architecture allows learning the statistical structure of a changing environment" for consideration by *eLife*. Your article has been reviewed by 2 peer reviewers, and the evaluation has been overseen by a Reviewing Editor and Aleksandra Walczak as the Senior Editor. The following individuals involved in review of your submission have agreed to reveal their identity: Rami Pugatch (Reviewer #1); Guillaume Lambert (Reviewer #2).

Essential Revisions:

1) How would results change if the threshold-linear model is replaced by a Hill function? Some discussion is currently in the SI. More discussion is needed in the main text of how all results would change if a hill function is used throughout.

2) A unified compact discussion of self-activation, non-linearity and regulator excess in the main text.

3) Discuss the cost/benefits of the proposed learning mechanisms if the environment doesn't fit the internal model discussed here – e.g., requires more than a binary on-vs-off response or environmental statistics varies very fast.

*Reviewer #1:*

My recommendation is a to reorganize the paper according to the following comments:

Weaknesses:

The first weakness I wish to discuss is the oversimplification of the model. The authors assume a set of parameters D_i_ (demand) has to be tracked by a set of internal parameters P_i_ and define a quadratic penalty term, (which happens to be the only choice that is amenable to analytical treatment in control theory). To motivate this modeling approach they give two examples, one of which is the expression of a costly metabolic pathway that ideally should track the relevant nutrient. The problem with this model is the assumption that there is only one optimal solution P→=D→. This I think is too simplistic since (i) the dimension of the internal set of parameters π at least in the metabolic example is typically larger than the state parameters D_i_ (even if they represent the demand for a metabolite there are typically multiple alternative supply pathways) ; (ii) The possibility of multiple solutions P_1_,…, P_m_ which are equally optimal is ignored.

For example, absence amino-acid requires expression of proteins that can metabolize these amino-acids from an inorganic nitrogen source. However, presence of both amino-acids and inorganic nitrogen source poses a dilemma to the cell, and a solution that optimize say growth rate depends subtly on both the relative concentration of the inorganic nitrogen source vs. the concentration and availability of external amino-acids, as well as the protein investment in transporting vs. making the amino-acids from scratch. If these relative concentrations fluctuate, an optimal solution in terms of growth rate can either be to ignore the amino-acids, or to ignore the inorganic nitrogen source or to express a mixture of both and this is not captured by the model presented by the authors. And there are other cases where multiple alternative pathways in metabolism are present to satisfy a given demand.

As the great William Feller used to tell his students, the best in science consists of the general embodied in the concrete. I like the general result but I urge the authors to take Feller's challenge and find a concrete example to demonstrate the merit of their approach and use it in the paper.

This biological concreteness will also solve a second weakness I found in the paper, namely the form of the nonlinear response used by the authors in the main text and in preparing the main figures (to my understanding). The nonlinearity is of the form of 'threshold-linear' response. This is not biologically feasible as there is always some upper saturation in the response. I expected to see the results presented in the main text in terms of a Michaelis-Menten (MM) or Hill function form which saturate both at low and at high concentrations. Although such cases were analyzed in the SI, it wasn't clear to me whether the main conclusions would change if such functions were used instead throughout the paper.

Since the author emphasize the role of (i) excess of regulators, (ii) self-activation, and (iii) non-linearity of the regulators, I think it is appropriate to analyze their importance in the main text. For some reason the role of self-activation was deferred to the SI.

Also, I would have liked to understand their relative importance (with realistic MM or Hill non-linearity) keeping in mind that proteins are costly for the cell to make, so a minimal excess is perhaps better.

Finally, I would expect a discussion of an hypothetical experiment, where such excess is deliberately removed by say gene deletion by the experimenter – what would be the simplest experiment that will ascertain the effect you predict be? Will a simple up-shift / down-shift will do the job?

Another more technical weakness is the statistical model being a Gaussian process. The authors claim their result is more general so I wonder how their result change for processes without time scale or to processes with power-law correlations?

A weakness in the analysis of the benefit of having such mechanisms for 'learning' the statistics of the environment requires some attention. If the environments statistics changes rapidly, or if the environment poses more complex challenges that require a response which is not all-or-none I would expect a reduction in the benefit of such a mechanism. An example that was discussed, is the case where the environment statistics varies too fast compared to the time-scale for learning a new environment statistic. It would be nice to see a simple summary of when the authors predict such a system will be advantageous for cells, considering e.g. the excess cost of building and maintaining such machinery. This naturally leads to a discussion of possible experimental verification as mentioned before.

More importantly, if we consider an ensemble of cells with such control mechanisms, their mutual interaction can lead to complex dynamics, and it is not clear if in such a setting where each cell response to the global change caused by all other cells, there is still benefit for learning the statistics which now become intertwined with the behavior of the average of all cells.

Finally, I find a weakness in the modeling of the "nominal" product feedback inhibition (PFI). It is not fair to compare to a naïve single-step PFI since actual PFI's are not single step and there is much work on the topic, see in particular the set of papers by Ned Wingreen from Princeton and his collogues.

I like the paper, and I think my comments benefits the paper. I also understand that doing it all over again might require too much effort, so I leave it to you and to the editor to think what it the best way to address my concerns. Perhaps (hopefully) some of my comments are already taken care of, and the only change is to better explain what you already did. I apologize in advance if I misread or misunderstood parts of the paper and because of that made an unnecessary comment.

I suggest you take a look at the following work:

Goyal, Sidhartha, et al., "Achieving optimal growth through product feedback inhibition in metabolism.". PLoS Comput Biol 66 (2010):, 6, 6, e1000802. Web and the PRL they cite.

I also vaguely recall Stanislas Leibler had a PNAS paper on learning with Bin Kan Xue as first author, perhaps it is relevant too.

I too had a paper that is somewhat relevant (That Kobayashi which you cite, cites in his paper) --- https://arxiv.org/abs/1308.0623

Please do not cite it unless you find it relevant, I only mention it because after reading your work I realized that I can use your method to test my formula..

Regarding the prediction issue (that such a system predicts the future and respond to the prediction which it constantly updates). This sound to me a lot like model predictive control. So I think a comparison with model predictive control literature might be in place. In particular, a clear discussion of how do you experimentally differ between non-predictive to predictive control schemes, given that most if not all of the internal variables are not measured in a typical experiment seems necessary.

*Reviewer #2:*

Overall, I recommend the publication of this manuscript as long as the authors address the following minor comments and suggestions.

1. A piecewise linear model for the regulator activity like the one presented in Eqn. 2c and 4c may not be biologically realistic. Although it is clear that the linear model was chosen because of the ubiquity of linear activation functions in neural networks, the authors may need to better justify why they chose a linear function, or whether it matters at all what the activity function looks like. Indeed, the authors do look into using a Hill-like function to describe the regulation of biochemical activity in section S8.2, it would be illuminating to emphasize this observation and discuss the role of choosing a specific activity function.

2. Line 389: "...but maintaining such a perfect regulatory system would presumably be costly;". Please explain why or provide a citation supporting this claim.

3. Line 777: (typo) "...acts as a senor"

Fig. 1A: Not sure what this panel is trying to show (even from the figure caption). Perhaps show a third panel above the other two where the information about the exterior is removed but the cell "sees" the same environmental composition?

Fig. 2C: The subscripts for a_1_ and a_2_ are missing.

Fig. 2E: It is unclear what the x-axis and y-axis labels are. X = and Y =

Fig. 3H: the "H" label for panel 4H is not there.

Fig. 3I+J: It is unclear how the regulator levels map back to the environment. Perhaps show a small ellipse with the correct orientation alpha (ie. -60, +30, -30, +60) in each sector, at the top of the graph, with the corresponding magnitude of each vector \sigma?.

Fig. 3J: Similarly, the main message of this panel is not very clear: several lines overlap and it is difficult to track a single one over time. Split that panel into 5 graphs arranged vertically, each showing the trace of a single regulator?

Fig. 4: Panel labels "A" and "B" are missing

---

## [Author Response]

Essential Revisions:1) How would results change if the threshold-linear model is replaced by a Hill function? Some discussion is currently in the SI. More discussion is needed in the main text of how all results would change if a hill function is used throughout.

This is indeed a very important point. To exhibit the generality of the learning behavior we describe, we used a minimal model which allowed us to characterize the minimal set of key ingredients without obscuring them by the (inevitably large) parameter space of any realistic implementation. However, the fact that the behavior is retained in a realistic implementation is, of course, a crucial part of the argument. Relegating this discussion to the supplement was a poor decision.

A literal interpretation of the reviewers’ suggestions here might have been to regenerate the plots of Figure 3 & Figure 4 using a Hill function instead of the threshold-linear model. With an appropriate choice of the Hill parameters, this could be done so that the panels barely change (for reasons explained in SI lines 663-670). However, we felt that doing so would have addressed only the letter of the comment, not the spirit; in fact, including such plots could almost be construed as misleading. Our minimal model includes several simplifications. Adding figures with “partial realism” while retaining other simplifications would implicitly suggest that the form of nonlinearity is the biggest concern one might have; but in fact, the simplistic treatment of regulator self-activation in Eq. (4c) is no less important.

The questions raised by the reviewers – namely, robustness of the behavior to the simplifications of our model – are issues we thought a lot about while writing the paper, and we felt that any claims about generalizability of the behavior must be validated in a reasonably realistic setting – with not only Hill-shaped activation curves, but also a biochemically plausible self-activation. We completely agree that this discussion belongs in the main text, not the supplement. Thus, to address the reviewers’ concerns, we added a figure 6 to the main text (“Realistic implementations”, which includes two panels that were previously in the supplement), and greatly expanded the corresponding section of Results (lines 341-374; see also lines 174-177).

2) A unified compact discussion of self-activation, non-linearity and regulator excess in the main text.

We have merged the relevant material, previously fragmented between the main text and the SI, into a single section of the main text (revised Figure 5; lines 305-329).

3) Discuss the cost/benefits of the proposed learning mechanisms if the environment doesn't fit the internal model discussed here – e.g., requires more than a binary on-vs-off response or environmental statistics varies very fast.

To address this point we, first, changed Figure 1A, replacing the (unclear) original cartoon by a concise summary identifying the context where the problem we are considering / the mechanism we are presenting are relevant (new Figure 1A and its legend; also lines 45-48). Second, the revised manuscript identifies the control theory calculation (and thus our findings of near-optimality) as being model-specific; of biological relevance is the ability of the circuit to learn correlations of fluctuations, which is not specific to the Gaussian model (lines 235-238). Additional suggestions from Reviewer #1 related to this point were incorporated in lines 88-94 and 266-268.

Reviewer #1:My recommendation is a to reorganize the paper according to the following comments:Weaknesses:The first weakness I wish to discuss is the oversimplification of the model. The authors assume a set of parameters D_i_ (demand) has to be tracked by a set of internal parameters P_i_ and define a quadratic penalty term, (which happens to be the only choice that is amenable to analytical treatment in control theory). To motivate this modeling approach they give two examples, one of which is the expression of a costly metabolic pathway that ideally should track the relevant nutrient. The problem with this model is the assumption that there is only one optimal solution P→=D→. This I think is too simplistic since (i) the dimension of the internal set of parameters π at least in the metabolic example is typically larger than the state parameters D_i_ (even if they represent the demand for a metabolite there are typically multiple alternative supply pathways) ; (ii) The possibility of multiple solutions P_1_,…, P_m_ which are equally optimal is ignored.For example, absence amino-acid requires expression of proteins that can metabolize these amino-acids from an inorganic nitrogen source. However, presence of both amino-acids and inorganic nitrogen source poses a dilemma to the cell, and a solution that optimize say growth rate depends subtly on both the relative concentration of the inorganic nitrogen source vs. the concentration and availability of external amino-acids, as well as the protein investment in transporting vs. making the amino-acids from scratch. If these relative concentrations fluctuate, an optimal solution in terms of growth rate can either be to ignore the amino-acids, or to ignore the inorganic nitrogen source or to express a mixture of both and this is not captured by the model presented by the authors. And there are other cases where multiple alternative pathways in metabolism are present to satisfy a given demand.

The biological question of determining the optimal behavior in a fluctuating environment is subtle. The optimal behavior may indeed not be unique, or even well-defined (optimal given what constraints?) The challenges faced by real organisms are certainly vastly more rich than our simplistic setup.

Nevertheless, generically, the most appropriate behavior will depend on statistics of fluctuations; and any organism attempting to make use of this would at a minimum need (1) a mechanism for sensing these statistics. Moreover, such a mechanism would (2) need to be simple enough to have plausibly evolved and be retained by evolution. Our minimal model allows us to focus on these two basic challenges.

In light of this, we agree with the reviewer that introducing our setup as a way to "model the general challenges of surviving in a fluctuating environment" in the original manuscript was poor phrasing. Our aim was to show that a simple circuit is capable of learning environment statistics, in a setup that could have plausibly benefited fitness. For this aim, we believe the simplifications of our setup (such as a unique optimum) are appropriate, and we added a paragraph (lines 88-94) to explicitly address this important point.

As the great William Feller used to tell his students, the best in science consists of the general embodied in the concrete. I like the general result but I urge the authors to take Feller's challenge and find a concrete example to demonstrate the merit of their approach and use it in the paper.

We certainly do agree with the sentiment. The metabolic motivation of the problem (line 128-132, lines 385-388 in discussion), the osmotic example (line 83-84) and the possible relation to two-component system architecture (lines 375-381, 395-400) represent our best attempt to include plausible concreteness without straying too far from the domain of our expertise. For example, we were quite intrigued by a possible implementation of our scheme in the context of protein synthesis and tRNA regulation, but we worried our distance from the subject would make it too cartoonish and naive. Similarly, our discussion of the two-component systems goes as far as we were comfortable taking it after a literature review and conversation with an expert (M. Goulian in acknowledgments). We therefore opted for explaining the idea in a theoretical language, explicitly separating it from examples. While a concrete and realistic example would have helped with making the message more broadly accessible, the fact that a reader like yourself saw possible parallels with e.g. the regulation of ribosome production -- a topic we ourselves did not consider -- appears to show that we did overall succeed at our intention.

This biological concreteness will also solve a second weakness I found in the paper, namely the form of the nonlinear response used by the authors in the main text and in preparing the main figures (to my understanding). The nonlinearity is of the form of 'threshold-linear' response. This is not biologically feasible as there is always some upper saturation in the response. I expected to see the results presented in the main text in terms of a Michaelis-Menten (MM) or Hill function form which saturate both at low and at high concentrations. Although such cases were analyzed in the SI, it wasn't clear to me whether the main conclusions would change if such functions were used instead throughout the paper.

This is an excellent point; please see the “essential revisions” section above (point #1) for a detailed list of changes we made to address it. For specifically the point on upper saturation in the response, see the edit in lines 667-668.

Since the author emphasize the role of (i) excess of regulators, (ii) self-activation, and (iii) non-linearity of the regulators, I think it is appropriate to analyze their importance in the main text. For some reason the role of self-activation was deferred to the SI.

Another excellent point; thank you. For the list of edits, see “essential revisions” section, point #2.

Also, I would have liked to understand their relative importance (with realistic MM or Hill non-linearity) keeping in mind that proteins are costly for the cell to make, so a minimal excess is perhaps better.

For relative importance: none of the three ingredients are sufficient on their own (Figure 5); in that sense, all are equally important. For minimal excess being better: indeed! The main feature of our scheme that (hopefully) makes it of plausible relevance to real living systems is that in the presence of nonlinearity & self-activation, a single additional regulator can already provide substantial benefit. We added a more nuanced discussion of these points (new panel Figure 5B; lines 317-329, especially 328-329).

For the purposes of this work, we postulated that sensing/learning fluctuations was desirable, and asked if it was possible. In any specific context, the full cost-benefit analysis would indeed weigh the benefit against the cost of making the necessary regulatory machinery. We did not perform this analysis; this is acknowledged in line 269 stating we use CIP as only a “proxy” for specifying the maximum tolerable cost.

Finally, I would expect a discussion of an hypothetical experiment, where such excess is deliberately removed by say gene deletion by the experimenter – what would be the simplest experiment that will ascertain the effect you predict be? Will a simple up-shift / down-shift will do the job?

One feature of the mechanism we described is that the excess regulators would appear fully redundant / dispensable in any static environment: the fitness penalty from their deletion would only manifest itself in a complex fluctuating environment, as a failure to learn. Of course, there are a great many functions whose benefits are masked in the simplified lab environment, so this feature is hardly a decisive identifying criterion. We added a discussion of a hypothetical experiment that could detect specifically the predictive nature of a regulatory scheme (lines 631-642). Our hope with this paper is that experts in any given system are better positioned to map the general idea onto their domain than we could ourselves.

Another more technical weakness is the statistical model being a Gaussian process. The authors claim their result is more general so I wonder how their result change for processes without time scale or to processes with power-law correlations?

Another good point; please see our response to the “essential revisions”, point #3.

A weakness in the analysis of the benefit of having such mechanisms for 'learning' the statistics of the environment requires some attention. If the environments statistics changes rapidly, or if the environment poses more complex challenges that require a response which is not all-or-none I would expect a reduction in the benefit of such a mechanism. An example that was discussed, is the case where the environment statistics varies too fast compared to the time-scale for learning a new environment statistic. It would be nice to see a simple summary of when the authors predict such a system will be advantageous for cells, considering e.g. the excess cost of building and maintaining such machinery. This naturally leads to a discussion of possible experimental verification as mentioned before.

To address this point, we made a new Figure 1A; the legend specifies that “In this work, we consider an environment undergoing epochs that differ in their fluctuation structure. Epochs are long compared to the physiological timescale, but switch faster than the evolutionary timescale.” The reviewer is correct in saying that, in principle, the discussion of the benefits would need to be offset by the cost of building/maintaining the relevant machinery. We did not perform this analysis here, and do not know how to do it in a way that is not system-specific; this limitation is acknowledged in lines 267-269.

More importantly, if we consider an ensemble of cells with such control mechanisms, their mutual interaction can lead to complex dynamics, and it is not clear if in such a setting where each cell response to the global change caused by all other cells, there is still benefit for learning the statistics which now become intertwined with the behavior of the average of all cells.

This is a very interesting comment. We edited the framing of the problem to specify we consider the fluctuations structure of externally imposed conditions (lines 82, 94). However, as the reviewer points out, the organisms also influence their own environment. Whether this fact makes the problem more relevant or less relevant is something that would be interesting to think about. Still, we feel that such a discussion lies outside of the scope of this manuscript. (However, we added a mention to lines 91-92.)

Finally, I find a weakness in the modeling of the "nominal" product feedback inhibition (PFI). It is not fair to compare to a naïve single-step PFI since actual PFI's are not single step and there is much work on the topic, see in particular the set of papers by Ned Wingreen from Princeton and his colleagues.

This is a good point. Although we keep comparing the performance of our scheme with product feedback inhibition, our intention is not to claim that this scheme is “worse” than ours; we simply use it as an example of a circuit that manifestly does not learn input statistics, in order to show that our circuit does. Our simplified treatment of PFI is not unfair to this scheme: if anything, our results actually reinforce prior findings (Pavlov et al., 2013, Goyal et al. 2010) that despite its simplicity, PFI is a remarkably efficient scheme -- in our Figure 4A, up until the separation indicated by the arrow, the SEPI curve is in fact optimal given its CIP! Our scheme is not generally better at the same CIP (note that before the arrow, our performance is worse than the blue “SEPI” curve), it just “knows” to upregulate the CIP to a larger value when fluctuations become large enough to activate the “learning”, and therefore performs better in some parameter regimes.

We added a paragraph to the main text stating the above (lines 276-282); and both references.

I like the paper, and I think my comments benefits the paper. I also understand that doing it all over again might require too much effort, so I leave it to you and to the editor to think what it the best way to address my concerns. Perhaps (hopefully) some of my comments are already taken care of, and the only change is to better explain what you already did. I apologize in advance if I misread or misunderstood parts of the paper and because of that made an unnecessary comment.

Thank you for your careful read of our manuscript and for the valuable suggestions that helped improve the work.

I suggest you take a look at the following work:Goyal, Sidhartha, et al., "Achieving optimal growth through product feedback inhibition in metabolism.". PLoS Comput Biol 66 (2010):, 6, 6, e1000802. Web and the PRL they cite.I also vaguely recall Stanislas Leibler had a PNAS paper on learning with Bin Kan Xue as first author, perhaps it is relevant too.I too had a paper that is somewhat relevant (That Kobayashi which you cite, cites in his paper) --- https://arxiv.org/abs/1308.0623Please do not cite it unless you find it relevant, I only mention it because after reading your work I realized that I can use your method to test my formula..

Thank you for the pointers. We read them all with great interest. The Goyal et al., is particularly relevant and we should have included it from the start; we added this reference as well as another one that seemed the most relevant (Pavlov et al., 2013).

Regarding the prediction issue (that such a system predicts the future and respond to the prediction which it constantly updates). This sound to me a lot like model predictive control. So I think a comparison with model predictive control literature might be in place. In particular, a clear discussion of how do you experimentally differ between non-predictive to predictive control schemes, given that most if not all of the internal variables are not measured in a typical experiment seems necessary.

We added a paragraph commenting on this to the SI section on control theory (section “Model predictive control”; lines 618-649).

Reviewer 2:Overall, I recommend the publication of this manuscript as long as the authors address the following minor comments and suggestions.1. A piecewise linear model for the regulator activity like the one presented in Eqn. 2c and 4c may not be biologically realistic. Although it is clear that the linear model was chosen because of the ubiquity of linear activation functions in neural networks, the authors may need to better justify why they chose a linear function, or whether it matters at all what the activity function looks like. Indeed, the authors do look into using a Hill-like function to describe the regulation of biochemical activity in section S8.2, it would be illuminating to emphasize this observation and discuss the role of choosing a specific activity function.

This is a very good point. Please see the “essential revisions” section above (point #1) for a detailed list of changes we made to address it.

2. Line 389: "...but maintaining such a perfect regulatory system would presumably be costly;". Please explain why or provide a citation supporting this claim.

Done (line 268).

Fig. 1A: Not sure what this panel is trying to show (even from the figure caption). Perhaps show a third panel above the other two where the information about the exterior is removed but the cell "sees" the same environmental composition?

Yes; we had tried many versions of this cartoon but in retrospect, it’s still confusing. In the revised manuscript, we replaced it by a concise summary identifying the context where the problem we are considering / the mechanism we are presenting are relevant. (New Fig. 1A)

Fig. 3I+J: It is unclear how the regulator levels map back to the environment. Perhaps show a small ellipse with the correct orientation alpha (ie. -60, +30, -30, +60) in each sector, at the top of the graph, with the corresponding magnitude of each vector \sigma?.Fig. 3J: Similarly, the main message of this panel is not very clear: several lines overlap and it is difficult to track a single one over time. Split that panel into 5 graphs arranged vertically, each showing the trace of a single regulator?

This was an excellent suggestion! We followed the reviewer’s advice to improve the labeling in these two panels, and find them to be much improved. (Updated panels 3I, 3J.)

Fig. 3H: the "H" label for panel 4H is not there.Fig. 2C: The subscripts for a_1_ and a_2_ are missing.

This may have been due to issues with PDF rendering on the reviewer’s machine; we do see these labels correctly on ours. If the paper is accepted for publication, we will make sure all labels are correctly displayed in the proofs.

3. Line 777: (typo) "...acts as a senor"Fig. 2E: It is unclear what the x-axis and y-axis labels are. X = and Y =Fig. 4: Panel labels "A" and "B" are missing

Corrected; thank you for pointing these out.